# Changes in phenology mediate vertebrate population responses to temperature globally

Phenotypic responses to climate affect individual fitness, but the extent to which this translates into effects on population dynamics remains poorly understood. We assemble 213 time series on phenotypes and population sizes of wild vertebrates globally and match them with local climate data. Our meta-analysis shows that morphological traits are mostly climate insensitive. However, phenology is earlier in warmer-than-average years, which contributes positively to population growth in most species. At lower latitudes, temperature has weaker effects on phenology but stronger direct negative effects on population growth, likely because these populations are less capable of tracking climate via plasticity. Variation in the phenology-mediated effect of temperature on population growth cannot be explained by latitude, generation time, migratory mode, or diet. This suggests that simple relationships between species characteristics and population responses to warming may not occur in nature. Instead, we may need to embrace ecological complexity by considering local-scale predictors that capture intra-specific variation.

Phenotypic traits are one aspect of biodiversity affected by ongoing climate change[1–5]. For example, the timing of recurring biological events such as reproduction and migration (phenology) is typically advancing with warming temperatures across taxa[1,3]. Similarly, morphology (e.g., body mass, size or shape) responds to climate change, although the directions of responses are less uniform than those of phenological responses[2,4–8]. Climate change has also led to population declines and increased risk of extinction[9,10]. Importantly, population responses are not independent of trait responses to climate variation, because individuals may adjust to changing climates by altering phenotypic traits (phenotypic plasticity), which, in turn, would enable population persistence (Fig. 1a)[11]. Studies that consider contributions of traits when assessing population responses to climate are rare and focus mainly on single populations or species[6,12,13]. Such studies highlight that phenotypic responses are key to understanding the mechanisms behind climate effects on populations[14–17]. However, little is known on how commonly traits mediate population responses to climate variation and allow for population persistence across species. Yet, such knowledge is crucial for the field of population ecology, as well as that of functional ecology, which relies heavily on the assumption that traits have direct population consequences and thus can serve as proxies to reflect community composition[18].

Earlier timing of biological events in warmer years is associated with fitness benefits on average[19] but it may also have costs by exposing individuals to extreme events, such as cold spells[20,21]. Along with understanding how responses of traits to climate relate to individual fitness, it is essential to assess their consequences for population growth, as changing population numbers are what ultimately determines the impact of climate on biodiversity. Although population growth is of primary importance to conservation, population sizes remain little studied in research focusing on phenotypic traits, such as phenology[22]. Indeed, whilst many studies have examined the consequences of trait change on single demographic rates (e.g., reproduction, survival) at the individual[16,23] or population level[4,6,21,24], few have examined the consequences of climate-driven changes in traits on population growth rate[8,11,12,14,25]. Studying the consequences for population growth rate is important because changes in fitness components (e.g. survival, reproduction) may not translate into changes in population growth rate[12,26] if demographic compensation is occurring[27,28]. Demographic compensation is a common

e-mail: radchuk@izw-berlin.de

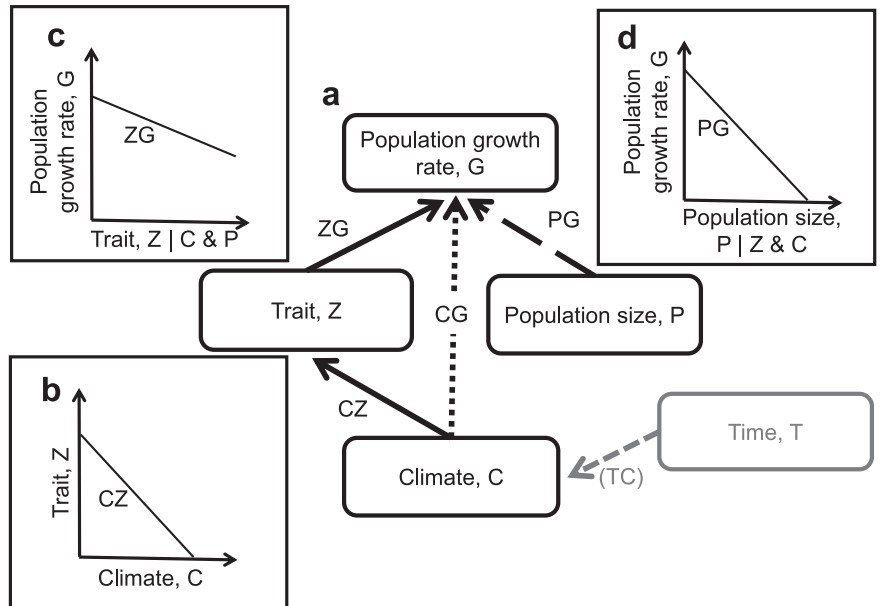

**Fig. 1 | A conceptual framework for assessing how phenotypic traits mediate effects of climate on population growth rate. a** General framework. Variables and paths shown in black were included in path analysis models: the effect of climate C on population growth rate G can be mediated by the considered trait Z (black solid line) and by other non-considered traits or direct effects (black dotted line, path "CG", read as "the effect of climate on growth"). Population size P is included to account for potential confounding effects (e.g., density dependence; black dashed line, path "PG"). Notation next to each arrow provides the name of the estimated path coefficient. Time was used outside of the path analyses to detrend the climate variable obtained with the sliding window analysis, prior to fitting the path analyses (dashed grey arrow from T to C, "TC"—see "Methods"). **b**–**d** show relations behind each path coefficient. The relations in **b**–**d** reflect our predictions for spring phenological traits. We also calculated the trait-mediated effect of climate on population growth rate ("CZG", i.e., the path from C to G that is mediated by Z and is found by multiplying the coefficients CZ and ZG for the two black solid paths from C to Z and Z to G) and the total effect of C on G (TotalCG, sum of the trait-mediated and other effects, see "Methods").

phenomenon[29] whereby population-level declines in a given demographic rate are offset by increases in another demographic rate[23,26,30]. Understanding the influence of climatic variation on populations therefore requires that we quantify the effects of climate-driven changes in phenotypic traits on population growth rates (Fig. 1). To assess how general such effects are across species, comparative analyses such as a meta-analysis are especially valuable[31].

In addition to quantifying the typical strength of trait-mediated effects of climate on population growth, another unanswered question is: in what type of species and geographic regions they will be strongest? Addressing these questions for vertebrates requires collating many decades-long time series of both phenotypic traits and population abundances that cover a range of climatic conditions for each study. Until recently, a sufficient number of such studies to fuel a rigorous comparative analysis was simply lacking. In situations of limited data, climate change ecologists often generalise the inferences from a group of well-studied species to predict the responses to climate for data-deficient species (i.e. most other species). These generalisations usually rely on species characteristics such as life-history traits, migratory mode, diet or latitude of occurrence.

Such species characteristics are likely to also explain trait-mediated effects of climate on population growth, but this has not been well investigated. We expect the effects of temperature on population growth rates, as mediated by changes in phenology, to be weaker in long-lived vertebrates compared to short-lived ones. This is because, in long-lived species, population growth rates are generally less sensitive to changes in reproduction and more sensitive to changes in survival[32,33]. Phenological shifts, such as changes in the timing of breeding or migration, are more likely to impact reproductive success than survival, and therefore should have a relatively weak effect on the overall population growth rate of long-lived species. Whether generation time is indeed a good indicator of how phenology mediates responses of population growth rate to climate

remains to be tested. Similarly, long-distance migrants advance their phenology to a lesser degree than residents because they have less reliable information about shifting environmental conditions at the breeding grounds[34–36]. However, we do not know whether migratory mode predicts how such phenological trait responses propagate further to affect population growth rate. Finally, phenological responses to climate may also vary with latitude, as the drivers of seasonality change from precipitation in the tropics to temperature at higher latitudes[1]. Consequently, tropical species are expected to respond more strongly to precipitation, whereas those at higher latitudes should respond more strongly to temperature changes. We expect the overall phenological responses to temperature to be stronger at higher latitudes[1] because of both faster warming[37–39] and higher sensitivity of species to warming at higher latitudes[39]. Whether latitude also explains variation in phenology-mediated effects of temperature on population growth remains unknown.

Responses of morphological traits to climate tend to be highly heterogeneous and no universal response has emerged so far[7,40,41]. A potential reason is that morphological traits are more heterogeneous than phenological traits, as they comprise both structural skeletal body size metrics (such as wing length) and non-structural traits (such as body mass). While body mass is a highly plastic trait that changes rapidly due to fat gains and losses, structural traits may be slower to change. Nonetheless, recent studies highlight that mean population values of body size metrics can vary considerably from year to year due to either reversible plasticity within a lifetime or developmental plasticity causing variation among generations[4,5,42]. Overall, the heterogeneity in morphological responses across species remains poorly understood, largely due to a scarcity of studies investigating how species characteristics mediate these[40]. Scarce research limits our ability to formulate expectations as clear as those for phenological traits and highlights the need for exploratory comparative studies to investigate how morphological responses to

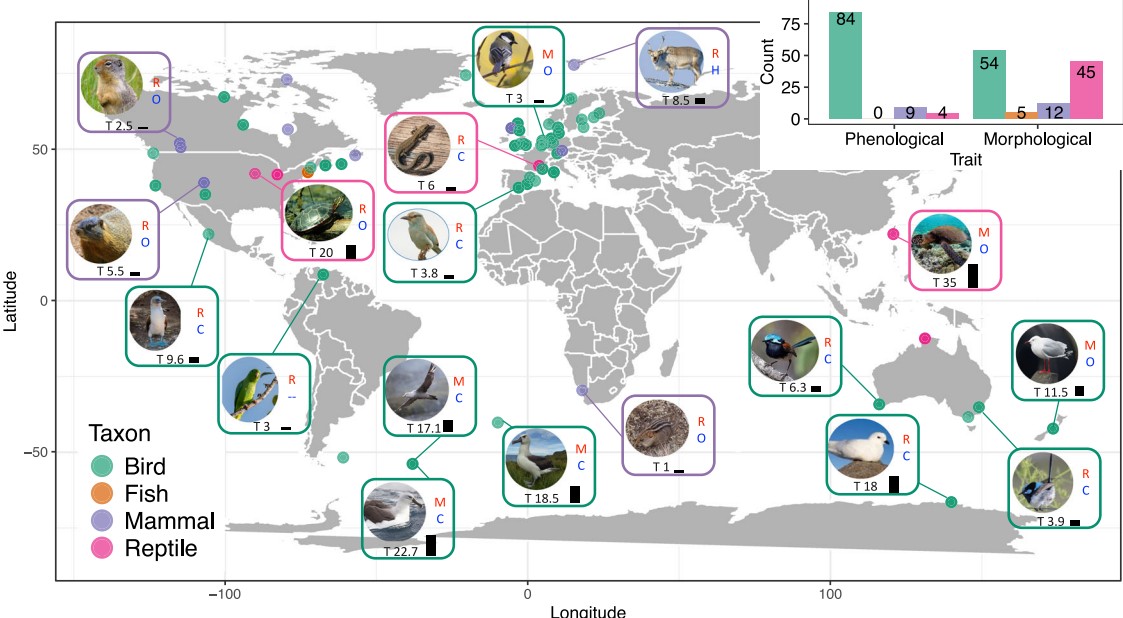

**Fig. 2 | Map of the studies, colour-coded for different taxa, with the number of studies per taxon shown.** A selection of studied species is shown, with each inset giving information for that species on its generation time (T, in years; also depicted by the black bar next to it), its diet (carnivore: C, herbivore: H, and omnivore: O) and whether the species is a migrant (M) or a resident (R). The inset shows the number of studies per taxon and trait category. Illustration credits for the species pictures taken from Wikipedia: Svalbard reindeer—Bjørn Christian Tørrissen, four-striped grass mouse—C.R. Selvakumar, silver gull—JJ Harrison, snow petrel—Samuel Blanc, northern giant petrel—Liam Quinn, green turtle—Brocken Inaglory, green-rumped parrotlet—Jam.mohd, Columbian ground squirrel—Martin Pot, red-winged fairy-wren—John Anderson, grey-headed albatross—John Harrison. Two species pictures were provided by the co-authors of this study: painted turtle (credit: FJ) and Atlantic yellow-nosed albatross (credit: SOp). The remaining pictures were taken from Pixabay (https://pixabay.com/photos/).

climate and their consequences for population growth rate depend on species characteristics.

Here, we collate long-term data from 213 studies on 73 wild vertebrates (birds, mammals, reptiles and fish) from around the globe (Fig. 2). Each study consists of time series of annual mean population phenotypic traits values and population sizes recorded for a unique combination of the species, trait and location. We focus on phenology and morphology as two categories of phenotypic traits often measured and frequently reported to play a role in climate responses[1–8,40,43–47]. We investigate the effects of two climate variables, temperature and precipitation, because these represent the main components of climate change[48] and are widely reported to affect phenology[1,3] and morphology[2,4–8,26,40,43–47]. In all analyses we consider year-detrended temperature and precipitation (i.e. residuals) to avoid potential spurious effects caused by other environmental temporal trends[49,50]. We use path analysis to (1) quantify the trait-mediated effects of climate on population growth rate (Fig. 1a), and meta-analyses to (2) assess how general those trait-mediated effects are across the studies and to (3) identify which type of species and regions exhibit the strongest trait-mediated effects of climate on population growth rate (using migratory mode, diet and generation time as explanatory species characteristics, and latitude to explain geographic variation among locations). Our study represents the first comprehensive assessment of the importance of phenotype-mediated population responses to climate variation across vertebrates, enabling us to test for general patterns that are hard to detect in more geographically- or taxonomically-focused studies.

## Results

Our systematic literature review resulted in 116 relevant studies focusing on morphology and 97 studies focusing on phenological traits. Median duration of morphological studies was 14.5 and of phenological studies 25 years (Supplementary Fig. S1). Roughly half of the studies focusing on morphology measured body mass (56%), with the other half recording different body size metrics (e.g. snout-vent length: 25%, body length: 5%, tarsus length: 4%). Most of the studies on phenology recorded onset of breeding (74%), followed by first egg laying date (8%), arrival date (6%), parturition date (4%), traits related to rutting (4%) and a few other phenological traits represented by very few studies. Most of the phenological studies (75%) focused on spring events such as return of migrants to breeding grounds, onset of breeding, and parturition dates. The dataset was dominated by studies on birds (65%), followed by reptiles (23%) and mammals (10%), with only a few studies on fish (2%, Fig. 2). The majority of data stem from the northern hemisphere, particularly Europe and North America (Fig. 2), as is the case with many recent global meta-analyses of ecological time-series[1,19,51].

### Conceptual framework

For each study, we assessed the effects of trait changes across years, associated with each climate variable, on interannual variation in population growth rate (henceforth 'G') using path analysis. We followed the framework of McLean et al.[11] that reflects the general expectation that the effects of climate variation on traits have consequences for species demography, such that the effect of climate C on G is expected to be mediated (at least in part) by the considered trait Z (Fig. 1a). Additionally, the effect of climate variation on G via all other non-considered traits is included as 'CG' (Fig. 1a, black dotted line) and hereafter referred to as the 'direct' (i.e. not mediated by the focal trait) effect of climate on G (though note that other traits may be involved in this pathway). We used the standardised path coefficients extracted from the path analysis to calculate the trait-mediated effect of climate on G, henceforth 'CZG', i.e. the change in G per unit change in climate C due solely to a change in the trait Z, where both changes are measured in standard deviation units (Fig. 1a, solid black line from C to G via Z). This was achieved by multiplying the path coefficient reflecting the effect of climate on trait (CZ) with the path coefficient reflecting the effect of trait on G[52] (ZG; Fig. 1a, Methods).

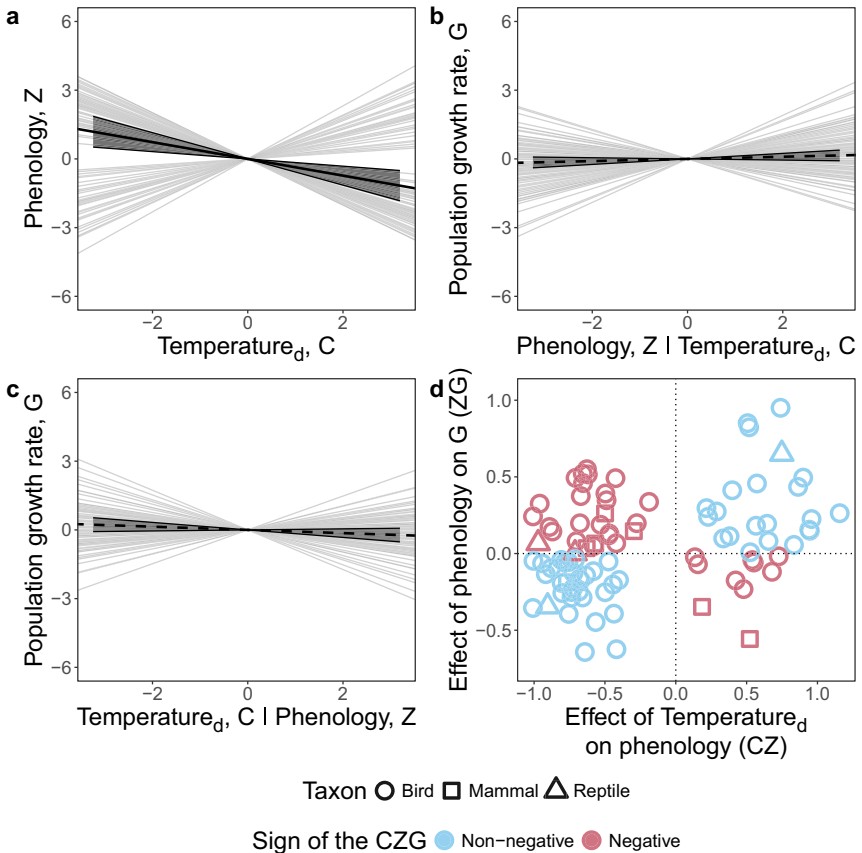

**Fig. 3 | Evidence that phenological responses to year-detrended temperature (temperature$_d$) propagated to population growth in most species.** Across studies, phenology was earlier in years warmer than average (**a**), the association between phenology and population growth rate conditional on temperature and population size was not significant (**b**), the direct effect of temperature$_d$ on population growth rate (mediated by all other traits but phenology) did not differ from 0 (**c**) and the proportion of studies with non-negative phenology-mediated effect of temperature$_d$ on population growth rate (CZG) was significantly higher than expected by chance (**d**). Grey thin lines in **a**–**c** show estimated slopes for each single study and black thick lines show the overall across-study effects. Solid thick lines demonstrate significant effects and dashed lines non-significant ones. The grey shaded bands around the black thick lines are the 95% confidence intervals. In **d**, studies with non-negative CZG are shown in blue and those with negative CZG—in mauve. The shape of the sign in **d** reflects the taxon.

We were specifically interested in the sign of the trait-mediated effect of climate on G: for each single study, we expect CZG to be zero or positive if the trait response to climate variation is adaptive. For example, in insectivorous birds, earlier egg laying in warmer years corresponds to a negative CZ (Fig. 1b). In this case, earlier breeding is expected to lead to an increase in G (negative ZG, Fig. 1c) if the optimal breeding time also advances, as trophic mismatches are then reduced[53]. The product of CZ and ZG (CZG) is thus positive. CZG will be close to zero if the trait responds to climate such that no effects on population growth rate are detectable. In other words, trait changes owing to adaptive phenotypic plasticity would allow accurate tracking of changing climate while avoiding possible detrimental effects of climate on population growth, such that ZG is close to zero (and hence so is CZG). On the contrary, we expect CZG to be negative for traits that show maladaptive responses, because in that case climate-driven trait changes would increase the gap between mean and optimum phenotype, rather than decrease it. For example, if breeding is delayed in warmer years, when in fact earlier breeding would improve fitness (e.g., facilitate better matching with an earlier food peak), then CZ (effect of temperature on phenology) is positive and ZG (effect of phenology on population growth) is negative, such that their product CZG is negative. In summary, if phenological or morphological responses to climate variation are adaptive, on average, across all studies in our dataset, we would expect the overall CZG to be non-negative, while negative CZG would suggest a maladaptive response.

The path coefficients obtained with path analyses from each study were used to fit meta-analytical models to assess the generality of responses across the studies. Meta-analytical models were fitted separately per combination of climate variable (temperature or precipitation) and trait category (phenology or morphology). Since there is evidence of phylogenetic structuring for phenological events[51] and evolutionary history may potentially shape trait-mediated effects of climate on populations, we accounted for phylogenetic relatedness in our meta-analytical models. Note that non-negative CZG on average across studies can be obtained not only when a majority of studies show positive or close-to-zero CZG, but also when effects of different sign are found across studies, e.g. some studies showing positive CZG (adaptive responses) and others showing slightly negative CZG (maladaptive responses).

## Climate effects are mediated by phenology

Our sliding window climate signal analysis provided strong evidence that phenological traits were sensitive to temperature, while most correlations between phenological traits and precipitation were likely spurious (see Methods and Supplementary Fig. S2). We thus henceforth focus on phenology-mediated temperature effects on G and use temperature$_d$ to refer to year-detrended temperature. Warmer years typically led to advanced phenology across studies ($\beta_{CZ} = -0.37$, Wald test: $\chi^2 = 12.5$, df $= 1$, $p \leq 0.001$), corroborating previous research[1,3]. However, we found considerable among-study heterogeneity in phenological responses to temperature$_d$ (CZ, Higgins $I^2 = 0.96$, $p < 0.001$),

with both advances and delays with warmer temperatures found for individual studies (Fig. 3a). The fact that none of the associations between temperature$_d$ and phenology (CZ) were close to 0 (Fig. 3a) was a result of using sliding window analyses to detect the best climatic windows over which temperature affects phenology (see "Methods"). This analysis identified the climate signals (models) best supported by the data; consequently, it was unlikely to select weak relationships between temperature$_d$ and phenology.

The across-study effect of phenology on G (ZG), after accounting for both the direct effects of temperature$_d$ (CG) and population size (PG), did not significantly differ from zero ($\beta_{ZG} = 0.05$, $\chi^2 = 1.64$, df = 1, $p = 0.2$), meaning that phenology was not associated with population growth rates across studies. The among-study heterogeneity in ZG was moderate ($I^2 = 0.48$, $p < 0.001$, Fig. 3b). Phenological responses to temperature$_d$ (CZ) were positively correlated with phenological effects on population growth rate (ZG, Pearson $r = 0.33$, df = 91, $p = 0.0013$), meaning that in studies that recorded positive effects of temperature$_d$ on phenology, the phenology was also positively associated with population growth rate (after accounting for the effects of CG and PG). Similarly, in studies that recorded negative effects of temperature$_d$ on phenology, the phenology was negatively associated with population growth rate.

Phenological responses to temperature$_d$ were often consistent with adaptive responses, as indicated by a significantly higher proportion of studies with non-negative CZG than expected by chance (binomial test, proportion of studies with either positive or zero CZG = 0.59, $p = 0.048$, Fig. 3d). After accounting for the phenology-mediated effects of temperature$_d$ on G, we found no 'direct' effect of temperature$_d$ on G across studies (Fig. 3c, $\beta_{CG} = -0.07$, $\chi^2 = 2.34$, df = 1, $p = 0.13$) and moderate among-study heterogeneity ($I^2 = 0.40$, $p = 0.01$). This suggests that the nett combined effects of temperature$_d$ on population growth via traits other than phenology are weak across studies, with a tendency towards negative direct temperature effects on population growth.

## Climate effects are not mediated by morphology

We found little support for morphological traits being climate-sensitive in vertebrates as correlations between both climate variables and morphological traits were mainly spurious, according to the sliding window analyses (see Methods, Supplementary Fig. S2 and Supplementary Note 2: Other combinations of climate variables and traits). This finding echoes previous research[8,43] on passerines, where little consistency was found across species in the timing and duration of windows over which precipitation was associated with body condition. In turn, this limited evidence for climate sensitivity in morphological traits in our dataset (CZ ≈ 0 in most studies) suggests that climate effects on G are unlikely to be mediated by morphology.

## Explaining among-study heterogeneity in climate effects

We investigated drivers of among-study heterogeneity in path coefficients by considering phylogenetic relatedness, differences among specific trait types and by adding species characteristics and latitude as predictors to the meta-analytical models. Phylogenetic signal $\lambda$ (Pagel's $\lambda$ is a scaling parameter related to phylogenetic signal[54]) was not distinguishable from 0 in the models focusing on phenological responses to temperature$_d$, nor in those focussing on phenological effects on population growth rate (Supplementary Fig. S3). Phylogenetic signal $\lambda$ was effectively 0 in all models fitted for 100 randomly drawn posterior vertebrate phylogenies, suggesting that evolutionary history does not explain CZ and ZG. However, phylogenetic structuring was evident for the phenology-mediated effect of temperature on population growth rate (CZG: $\lambda = 0.65$, min = 0.61, max = 0.68) and the direct effect of temperature$_d$ on G (CG: $\lambda = 0.62$, min = 0.56, max = 0.66).

The across-study patterns in phenological responses to temperature$_d$ (CZ), effects of phenology on G while accounting for climate and population size (ZG), phenology-mediated effect of temperature$_d$ on G (CZG) and direct effect of temperature$_d$ on G (CG) seemed to be predominantly driven by birds, which constituted most of our dataset (Supplementary Figs. S4 and S5). Since different types of phenological traits (e.g., onset of breeding, parturition date, oestrus date, and arrival date) may differ in their sensitivity to temperature, we tested whether the specific type of phenological trait could explain the heterogeneity in CZ. As expected, the among-study variation in phenological sensitivity to temperature$_d$ was explained by the type of phenological trait considered (Wald test: $\chi^2 = 19.6$, df = 8, $p = 0.012$, Supplementary Table S1). For example, the response to temperature$_d$ was strongest for spring phenological events, such as parturition date and onset of breeding, whereas the association of autumn phenological events (such as rutting date) with temperature was less clear.

We expected (see Introduction) that the considerable heterogeneity in phenological responses to temperature$_d$ (CZ; Fig. 3a) across studies would be partly explained by species characteristics such as generation time, diet (herbivore, omnivore and carnivore), and migratory mode (resident vs. migratory), and by latitude. The model including these effects was significantly better than the null model (Wald test: $\chi^2 = 17.2$, df = 6, $p = 0.009$; Supplementary Fig. S6). Absolute latitude tended to affect phenological sensitivity to temperature$_d$ ($\beta = -0.015$, $\chi^2 = 5.6$, $p = 0.018$; with the $p$-value threshold of 0.01, adjusted for multiple comparisons, Fig. 4a and Supplementary Table S2). Phenological responses to temperature were weak at lower and became stronger towards higher latitudes. Neither the model that explained among-study variation in ZG ($\chi^2 = 9.13$, df = 6, $p = 0.104$) nor the model that explained heterogeneity in CZG ($\chi^2 = 6.14$, df = 6, $p = 0.407$) was significantly better than a null model (Supplementary Tables S3 and S4). However, we found that the model explaining CG fitted the data better than the intercept-only model ($\chi^2 = 21.3$, df = 6, $p = 0.002$). The 'direct' effect of temperature$_d$ on G that remained after accounting for the phenology-mediated effect of temperature$_d$ on G was positively associated with absolute latitude, such that it changed from being negative at lower latitudes to around zero towards higher latitudes ($\beta = 0.008$; $\chi^2 = 14.3$, $p = 0.0002$, with the $p$-value threshold of 0.01, adjusted for multiple comparisons, Fig. 4d and Supplementary Table S5).

## Sensitivity analyses

Our analyses revealed a strong negative effect of population size on G (Supplementary Fig. S7)[12,55]. In line with the large magnitude of estimated effect sizes for population size, we also found that population size explained the largest proportion of variation in G compared to other predictors (see Supplementary Note 3: Variance partitioning in SEM). Furthermore, the probability that detected windows were spurious was not negligible for phenological responses to precipitation$_d$, and for morphological responses to both temperature$_d$ and precipitation$_d$ (Supplementary Fig. S2). However, the results of the meta-analyses were not qualitatively affected by inclusion/exclusion of (1) the less supported climate signals in the models and (2) population size as a covariate (Supplementary Figs. S8–S11). The probability of the temperature signals being spurious ($P_{\Delta AICc}$) was higher for morphological studies and declined with the study duration (Supplementary Fig. S12). The magnitude of the absolute CZ effects declined slightly with increase in $P_{\Delta AICc}$, so that the effects of climate on trait (irrespective of the trait category) tended towards 0 as $P_{\Delta AICc}$ approached 1 (Supplementary Fig. S13).

## Discussion

Advancement of phenology is a widely observed biological response to warming for most taxa[1,3]. Here, by using a large set of species and studies, we demonstrate that the phenology-mediated effect of temperature on population growth rate was on average non-negative across studies. Thus, spring phenology facilitates adaptive population

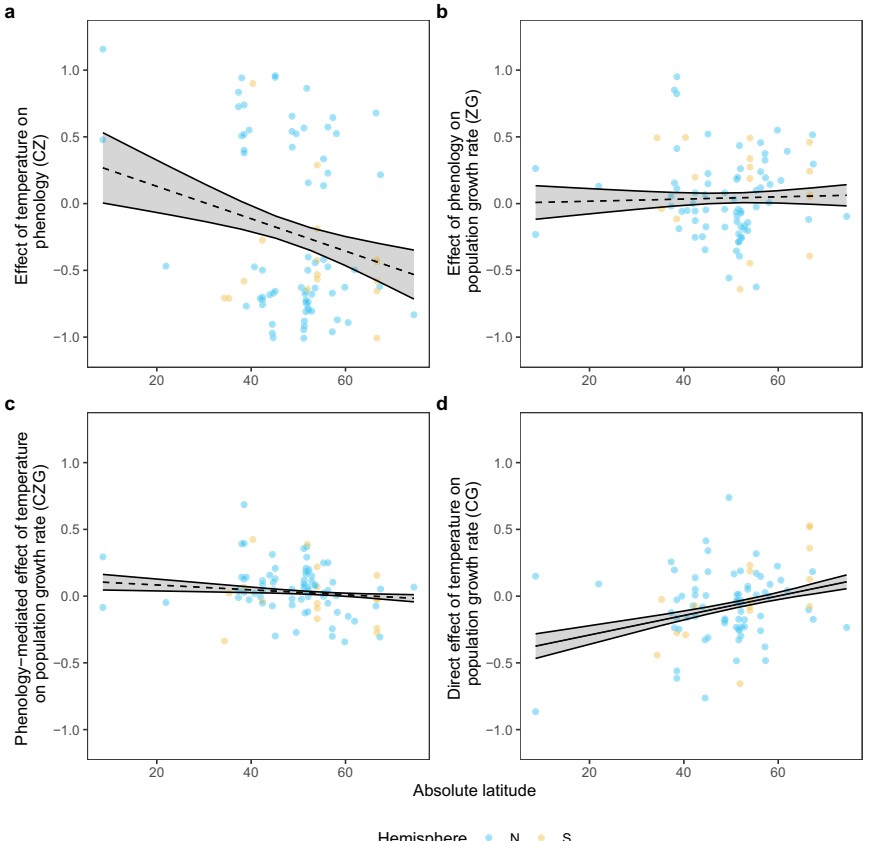

**Fig. 4 | Absolute latitude partially explains across-study variation in the path coefficients estimated using our conceptual framework (Fig. 1). a** The sensitivity of phenology to temperature$_d$ (CZ) tends to increase from the equator to the poles (two-sided Wald test: $\chi^2 = 5.6$, df = 1, $p = 0.018$, $\beta = -0.015 \pm 0.006$); The association between phenology and population growth rate (ZG, **b**) and the phenology-mediated effect of temperature$_d$ on population growth rate (CZG, **c**) do not change significantly with latitude (two-sided Wald tests: $\chi^2 = 0.1$, df = 1, $p = 0.787$, $\beta = -0.001 \pm 0.003$ and $\chi^2 = 0.14$, df = 1, $p = 0.285$, $\beta = -0.002 \pm 0.001$, respectively); **d** the effect of temperature$_d$ on population growth rate via all other non-considered traits ('direct' effect of temperature) switches from negative at the equator towards being close to zero and even positive at high latitudes (two-sided Wald test: $\chi^2 = 14.2$, df = 6, $p = 0.0002$, $\beta = 0.008 \pm 0.002$). The $p$-value threshold adjusted for multiple comparisons is 0.01. Data lines and shaded regions show model fits and ±1 standard deviation, respectively (solid line for the significant effects, and dashed line for non-significant effects). The points show the raw data. Although the models were fitted with grand-mean centred predictors, we here show the original values of absolute latitude, to aid interpretation.

growth responses to temperature in many vertebrate species found across much of the globe. However, we found high heterogeneity across studies, with a substantial number showing a negative maladaptive response, partly driven by type of phenological event. Our results are predominantly driven by birds, because the relevant data were mainly available for this taxon, despite our initial intention to address this question for vertebrates broadly. The finding that phenology typically mediates effects of temperature on population growth rate is particularly important for the field of functional ecology, which is founded on the assumption that traits have direct population consequences[18,56]. In most populations, the phenology-mediated effects of temperature on population growth rates were positive or zero, consistent with adaptive phenotypic responses. These responses are likely mostly explained by phenotypic plasticity, rather than microevolution, given the timescales involved, the fact that phenological traits are well-known to be plastic[57–63], and previous research demonstrating that detrending temperature, as was done here, can reveal evidence of phenotypic plasticity[50]. We identified latitude as a geographical variable explaining inter-study variation in both phenological responses to temperature and the effect of temperature on population growth rates via traits other than phenology.

Our estimated across-study effect of absolute latitude on the association between phenology and temperature ($-0.015 \pm 0.006$ S.D. trait change per °C per degree latitude) is in the same direction as that found by Cohen et al.[1] ($-0.005 \pm 0.0005$) and implies that phenology becomes more temperature-sensitive closer to the poles (Fig. 4a). Here we also showed that neither latitude (nor species characteristics) could explain the variation in either ZG or CZG, but variation in CG was associated with absolute latitude (Fig. 4). This "direct" effect of temperature on population growth switched from being negative in tropical regions to around zero at higher latitudes. One possible explanation for both latitude effects is that the reduced sensitivity of phenology to temperature in lower-latitude populations renders them more vulnerable to trophic mismatch. Another is that the traits of lower-latitude populations are in general less plastic and hence these populations have narrower thermal tolerance than higher-latitude populations[64–67] that typically experience higher intra-annual temperature variation.

The effects of species characteristics on the temperature sensitivity of phenology (CZ) and on phenology-mediated population responses to temperature (CZG) were non-significant, which may reflect low statistical power. Indeed, we had fewer data points than previous studies that focused purely on trait responses to climate[1,3], because addressing our research question required parallel time series on both population size and the trait for each population. Our dataset comprised 97 time series, which is not a small sample per se, but because we relied on previously published data, we had no control over the distribution of the species characteristics within this dataset.

Therefore, future extensions of the dataset aimed at covering a wider range of species characteristics would allow for better addressing of this particular question. Alternatively, species characteristics such as generation time and diet may truly be poor predictors of phenology effects on population growth. Intra-specific variation in trait responses to climate can be large, as demonstrated for phenological responses to temperature among populations in European songbirds[38]. Further, such intra-specific variation may be higher compared to interspecific variation[8] and this may also be the case here, as variances in phenological responses to temperature due to differences among species and among study locations were of similar magnitude (Supplementary Table S6). Taken together, among-study variation is likely to be better explained by predictors related to local environmental conditions describing the habitat of the population, rather than generic metrics that apply to the species as a whole. Indeed, habitat type explained within-species variation in phenological sensitivity to temperature in a study of two songbirds[38]. Similarly, altitude can also help explain differences in phenological responses to warming for populations that are in close proximity. Indeed, Uinta ground squirrels (*Urocitellus armatus*) at low elevations can emerge and access food much earlier (6–8 weeks in some cases) than their high elevation counterparts just 3 km away[68]. To clarify the role of species characteristics, future research should focus on increasing the sample sizes and diversity of species types considered, and on including intraspecific predictors (such as habitat type) in the analyses.

Our phylogenetically-corrected meta-analyses focusing on phenological responses to temperature (CZ) and on effects of phenology on population growth rate (ZG) did not detect strong phylogenetic signals (not distinguishable from 0). Similarly, a recent meta-analysis found that phenological shifts were structured phylogenetically only when running the analyses on the full dataset that spanned across species in 29 taxonomic classes, while the authors found no support for phylogenetic signal when re-running those analyses on four lineages (amphibians, birds, insects and plants) separately[51]. Most of our data on phenological responses comes from birds with only nine and four studies on mammals and reptiles, respectively. We thus may not have encompassed enough of the vertebrate tree to pick up the phylogenetic signal. The phylogenetic signals remained indistinguishable from 0 when we re-ran the analyses for birds and mammals separately (see Supplementary Note 4: Phylogenetic signal). Interestingly, we found that variation in the phenology-mediated effect of temperature on population growth rate was phylogenetically structured, as was variation in the direct effect of temperature on population growth rate, suggesting that evolutionary history may shape population responses to temperature. These findings are similar to the results by McLean et al.[8], who demonstrated that direct effects of temperature on population growth rate (that are not mediated by body condition, a focal trait in their analyses) were mainly due to among-species and not within-species variation.

We found no evidence for climate sensitivity of morphological traits (CZ ≈ 0), which implies that morphology does not mediate population responses to climate. Skeletal traits may be less plastic than non-skeletal morphological traits and thus the former may change more slowly (e.g., via evolution over thousands of years[69]) in response to climatic changes. As a result, skeletal traits exhibit less inter-annual variation compared to phenological traits, which impedes the detection of relationships between these traits and detrended climate variables. As changes in both morphology and phenology arise from changes in underlying physiological traits (e.g., gonadal development for laying date)[70], we expect that physiological traits would respond much faster than phenological or morphological ones and likely mediate many climate effects on populations. However, there are few long-term studies on such traits in the wild, as these typically must be done in the lab.

Our findings must be interpreted carefully due to limitations of the dataset, despite it being the largest of its kind linking climate-induced changes in phenotypes to population dynamics. Our dataset was assembled from published studies and thus reflects biases present in the current research. Geographically, our dataset is heavily biased towards the northern hemisphere, especially Europe and North America (Fig. 2). This bias, unfortunately, is a norm for all similar meta-analyses[1,19,51], highlighting the need for more research in the Global South. Future studies covering the latitudinal gradient more systematically should aim to validate our findings. Further, the studies in our dataset mainly focus on birds and spring events, and more studies are needed on other taxa and autumn phenology to obtain a more complete understanding. Finally, temperature and precipitation were not available at the same resolution globally, and the coarser resolutions for the Americas and Africa might have introduced some unwanted heterogeneity.

In conclusion, we found substantial variation among vertebrates in trait responses to temperature and could partially explain this variation by latitude. We demonstrate that phenological responses to temperature propagate to affect population growth on decadal timescales. The fact that phenology mediates population responses to temperatures across many vertebrates highlights the importance of incorporating phenotypic traits more systematically into research focusing on climate effects on populations, whether based on observational, experimental or modelling studies. Our study provides much-needed estimates of both climate effects on traits and trait effects on demography, which will facilitate parameterisation of mechanistic population models across vertebrates[71]. The phenology-mediated effect of temperature on population growth rates was positive or zero in the majority of the studies, consistent with adaptive responses. Though we showed that phenology mediates population responses to temperature, we could not explain variation in such responses with any of the tested species characteristics. This finding is consistent with the increasing body of research showing that species characteristics are weak predictors of, for example, species range shifts[72,73] and plant demography[73]. Whilst being able to identify species characteristics that are informative of population responses to climate would be highly desirable from the standpoint of conservation biology, our global-scale comparative study spanning four vertebrate taxa indicates that we may not find such simple relations in nature. Instead, we may need to embrace ecological complexity by studying multiple phenotypic traits and considering predictors that capture intra-specific variation.

## Methods
### Biological data
Our dataset was compiled by combining an existing global multi-species dataset[19] (Phenotypic Responses to Climate and data on Selection; PRCS) with a new systematic literature review. Studies were included if they reported both a time series of annual mean population trait (phenological or morphological) values (+SE), and a time series of population size estimates (based on a capture-mark-recapture study or a count of individuals or breeding pairs) of at least 9 years. Only quantitative traits were considered, and only studies investigating the impact of a climatic variable on traits were included.

We conducted a systematic literature review using Web of Knowledge (now called Web of Science; search conducted in April 2019) including the key words for climate, phenotypic traits, demographic rates and population size. We used the following key words for climate (climat * OR temperature OR precipitation OR weather), phenotypic trait (phenotyp* OR morphol* OR "body mass" OR "body size" OR phenol* OR "emerg* date" OR "arriv* date" OR "breed* date" OR "laying date"), demographic rate (demograph* OR "demographic rate" OR surviv* OR reproduc* OR fecundity OR "breeding success"), and population size ("population growth" OR "population dynamics" OR "population size" OR abundance). For taxa, we first used broad taxon names ("bird*" OR "mammal*" OR "arachnid*" OR "insect*" OR "reptil*" OR "amphibia*" OR "spider*" OR "fish"), then proceeded by using more

specific taxon names to find as many relevant studies as possible (e.g. for mammals we searched for (rodent* OR mammal* OR primate* OR carnivore* OR bat*)) (see SI Methods). The review was conducted within a larger research project (https://www.idiv.de/research/sdiv/working-groups/straitchange/), where we look at how the effects of climate on population growth rates are mediated by phenotypic traits and demographic rates (e.g., survival and reproduction). Therefore, we searched for studies reporting the time series of annual population-level trait values, demographic rates and population sizes, but in this study we used only the data on traits and population sizes.

The search returned 1124 abstracts, which were divided among eight researchers (VR, CVJ, GC, EM, TER, JC, SKS, and NMcL) who screened the abstracts to identify those that satisfied the above-mentioned criteria. This resulted in 197 scientific papers that were read in detail (by VR and CVJ), of which 60 papers were retained (Supplementary Fig. S14 shows PRISMA flow diagram). Wherever possible, we extracted the data from the papers directly, either from tables or by digitising the plots with WebPlotDigitiser[74]. Otherwise, we contacted the authors and asked them to share their data. The 82 studies (stemming from 13 papers) from the PRCS dataset originally did not contain data on population size. We therefore compiled the time series on population size by either extracting it from the papers ourselves or by contacting the authors of the original papers. We define 'study' as a unique combination of a species, location and trait (phenology or morphology). Our final dataset consisted of 213 studies extracted from 73 papers. Our studies cover four vertebrate classes: birds (53 species), mammals (10 species), reptiles (7 species) and fish (3 species; see Supplementary Data 1 and Fig. 2 for study sample sizes per trait category). Some papers contributed multiple studies to our dataset when multiple traits and/or species and/or locations were recorded. Morphological traits were represented by 116 and phenological by 97 studies.

## Identifying suitable climatic windows

The spatial resolution of the daily climate data ranged from 0.05 deg in Australia, to 0.1 deg in Europe and 0.25 in North America and 0.5 deg elsewhere (Supplementary Table S7). For each study, we extracted the daily climate data from the respective gridded datasets as the mean of the climate value in the grid cell that overlapped with the study location and the four neighbouring grid cells (von Neumann neighbours). Such spatial averaging was done to account for the possibility that home ranges of study species may be larger than a single grid cell. The climate conditions reflect conditions at breeding sites, and do not consider potential effects during migration or at wintering sites. In cases where study locations did not overlap with the available gridded data (some of the island populations: 83 studies, 39% of all studies), we used the closest grid located on the mainland. These climate data were potentially less precise, which we accounted for by adding a categorical variable 'Climate data quality' with values 'exact' and 'approximate' to the meta-analysis (see section 'Across-study inferences'). Importantly, most islands were <50 km offshore or focused on seabirds for which we used sea surface instead of air temperature.

We identified a period of the year (climatic window) during which the climatic variable best explained the studied trait by applying a systematic sliding window analysis using the R package *climwin*[75]. The analyses were conducted separately for precipitation and temperature, by fitting a linear model with Gaussian error distribution for each study. We used the studied phenotypic trait as the response variable. As temperature predictor we used the mean temperature over the climatic window, and as precipitation predictor we used the total sum of precipitation over that window. We included year as a quantitative covariate in the model, to avoid detecting spurious climate windows because both climate and other environmental drivers of traits may be changing directionally over time concurrently[49]. We weighed model residuals by the inverse of the squared SE of the annual trait values, to account for varying sample size across years. For our sliding window

analyses we assumed that all individuals in the population had the same climatic window (i.e., we treated windows as 'absolute'). We tested for all possible windows (see Supplementary Note 5: Climatic window durations) over a period of 2 years before a so-called reference day, the value of which was study-specific. The reference day was the latest date when the phenological event was observed or morphological trait was measured in the year over the study period. By using the latest day in the year when the phenological trait was observed over the study period we assured that no relevant windows was missed (Supplementary Fig. S15). The use of a 2-year period prior to the biological event ensured that the analyses captured carry-over effects of climatic conditions experienced by individuals previously. Our dataset includes diverse species, ranging from birds to mammals, reptiles and fish. For many of these species biological knowledge on what constitutes a relevant climatic window is unavailable. Therefore, the use of sliding window analyses allowed to assess the climatic window in a standardised way, systematically, across all studies and species. Our identified climatic windows for passerines correspond to the windows that are known to be important in driving egg-laying of these birds based on long-term studies, suggesting that our approach is valid. Our identified climatic windows may reflect proxies of the true underlying biological mechanisms. Our consideration of sliding windows up to 2 years before an observation may include potential climatic windows that are not very meaningful for short-lived species (those whose life span is <2 years). However, only five of the species (7%) in our dataset have generation time ≤2 years, justifying the use of the 2-year period to capture potential carry-over effects for most of the much longer-lived species in our dataset. To reduce computation time, the daily climate data were aggregated to weekly resolution (option interval = 'week' in *climwin*), as preliminary analyses showed similar outcomes for weekly and daily resolution (Supplementary Fig. S16). The median window duration across the studies was 3 and 2 weeks for precipitation and temperature, respectively (Supplementary Fig. S17).

To ensure that the climatic window selected as the best (as determined by the model's ΔAICc) was not a spurious result due to overfitting, we randomised the original data to remove any potential relationship between the climate and trait variables and refitted the model as explained above (for details see ref. 76). The randomisation procedure was repeated 200 times. The results from the randomised data were compared to those from the observed data by calculating the metric $P_{\Delta AICc}$, which we henceforth call "the probability of the climate signal being spurious" and which reflects the probability of obtaining the same results as those that would be produced by chance only. To account for the less accurate identification of the climate signal in some studies, we included the probability of the climate signal being spurious ($P_{\Delta AICc}$) as a covariate in the meta-analyses (see section 'Across-study inferences').

## Trait-mediated effects of climate on G

To assess the consequences of the effects that climate variables on traits have for the population growth rate (G, Fig. 1), we used the path analysis[52,77] framework proposed by McLean et al.[11], and further extended it to account for potential effects of density dependence (Fig. 1a), a common phenomenon in natural populations[12,55]. Path coefficients were estimated with Structural Equation Models (SEM[77]), an extension of path analysis that allows for non-independence in the data (e.g., autocorrelation). We fitted a single SEM to each study in our dataset, and separately for each climatic variable C. We calculated the annual population growth rate as

$$G_t = \ln\left(\frac{P_{t+1}}{P_t}\right) \qquad (1)$$

where $P_t$ and $P_{t+1}$ are the annual population sizes at year $t$ and $t+1$, respectively. Each SEM consisted of two regression models: one

model ("the trait model") with annual trait values (Z) used as response and climate variable (C) as a quantitative predictor (providing CZ, see Fig. 1); and the second model ("the population growth model") with population growth rate $G_t$ as response and climate variable ($C_t$), trait values ($Z_t$) and population size $P_t$ used as quantitative predictors (providing ZG, CG and PG, Fig. 1). Both linear models were fitted by generalised least squares, with Gaussian error distributions and identity links and included temporal autocorrelation by using a first-order autoregressive residual model structure. Additionally, for the "trait models" we weighted the residual variation by the inverse of the squared SE of the traits to account for differences in sample sizes among the years. Population size was used as a predictor in "population growth models" to account for the possible changes of population size over time that could have obscured the effects of climate C on G, which are the focus of our study. We fitted linear relationships between predictor and response variables, as linearity was supported by (1) visual inspection of the scatterplots per study and (2) mixed-effects models testing for non-linearity in the relations between (a) climate variable and trait values, and (b) trait values and population growth (see Supplementary Note 6: Testing for non-linearity in relations). SEMs were fitted with the R package *piecewiseSEM*[78]. For all analyses in this study we used R Statistical Software (v4.4.1[79]).

Yearly climate values from the best climate window could correlate with other unmeasured environmental variables that changed concurrently over time[49,80]. Therefore, to ensure that we focus on the changes in traits that are caused by the considered climate variable only, we first detrended our climatic time series[49]. For this, we fitted a linear model using the climate variable extracted from the best-identified climate window as a response variable and year as a quantitative predictor variable. The residuals of this model were then used as the explanatory climate variable in our SEMs (Supplementary Fig. S18 for methodological workflow of this study).

Prior to fitting SEMs, we z-transformed (subtracted the mean and divided by the SD of the time series) all variables to ensure that the extracted path coefficients were standardised. We also scaled the standard errors of traits by dividing them by the temporal SD in traits over the study period. The advantage of the standardised path coefficients is that they are unitless and can be compared among different variables (i.e., different traits: phenology vs. morphology; and different climate variables: temperature vs. precipitation).

SEMs for most of the studies satisfied the goodness of fit test according to Fisher's C statistics (Supplementary Fig. S19): the goodness of fit test was satisfied (i.e. *p*-value > 0.05) for 72% of phenological studies and 84% of morphological studies. Although goodness of fit was not satisfactory for a small fraction of the models, we still retained all the studies for subsequent meta-analyses because we wanted to assess the trait-mediated effects across the studies in a comparable way. So, the addition of some missing paths (that could potentially increase the goodness of fit for that fraction of the models), would have made those models not comparable with the others.

SEMs failed to converge for several studies resulting in sample sizes for meta-analyses being somewhat lower than the number of originally retrieved studies: 93 studies for phenological responses to temperature, 95 studies for phenological responses to precipitation, 109 studies for morphological responses to temperature and 115 studies for morphological responses to precipitation (Supplementary Fig. S7). The models constituting the SEMs explained a non-negligible amount of data variation (Supplementary Figs. S20 and 21), with the largest $R^2$ obtained for the "population growth models", i.e. models explaining G (median across all studies 0.47), followed by the "trait models" (median of 0.37).

We used the rules of path coefficients[52] to compute the trait-mediated effects of climate on G, and the total effects of climate on G.

Specifically, the trait-mediated effect of climate on G (CZG, shown with solid black arrows from C to G via Z in Fig. 1a) is calculated as:

$$CZG = CZ^*ZG \qquad (2)$$

where CZ denotes the path coefficient corresponding to the effect of climate C on trait Z and ZG denotes the partial path coefficient corresponding to the effect of trait Z on G after accounting for the effect of climate C and population size P. The total effect of climate on G (TotalCG) is then calculated as the sum of the trait-mediated effect (solid black line in Fig. 1a) and the climate effect via other unconsidered traits (dotted black line in Fig. 1a):

$$TotalCG = CZG + CG \qquad (3)$$

where CG corresponds to the effect of climate on G that is not mediated by the focal trait but acts via other traits not considered in the model. We obtained estimates of the trait-mediated and total effects of climate on G and their standard errors by non-parametric bootstrapping. In this bootstrap the trait-mediated and total effects of climate on G were calculated using Eqs. (2) and (3), respectively, but instead of using single estimates of each path coefficient as obtained with the SEM, we calculated these coefficients 10,000 times by drawing the estimates each time randomly from the normal distribution with the mean being the beta estimate for each single path coefficient and the variance being the standard error (SE) of this estimate. This bootstrap allows us to consider the uncertainty in the estimates of each path coefficient on the path of interest, so that the path coefficients that are more precisely estimated (with low SE), would have very similar estimates each time the bootstrap is run, whereas the path coefficients with large uncertainty (with high SE), will have their value vary widely between the bootstrap runs. We then summarised the results of 10,000 runs by taking median as our estimate of the trait-mediated (total) effect of climate on G, and by using the 0.025% and 0.975% quantiles as its 95% confidence intervals.

## Across-study inferences

To assess whether there are common patterns across the studies in how climate affects traits (CZ), traits affect G (ZG), and in trait-mediated effects of climate on G (CZG), we next analysed the standardised path coefficients from SEMs (Fig. 1a) with mixed-effects meta-analytical models with Gaussian error distributions. Separate meta-analyses were fitted for each climate variable and trait category, using each path coefficient that reflects the causal relation on the path diagram (e.g., CZ in Fig. 1a) as a response variable. To assess how general responses are across the species, we fitted intercept-only models that accounted for variation between studies, locations and species by including these terms as random intercepts. Additionally, these models also accounted for the phylogenetic relatedness by allowing the values of the random species effect to be correlated according to the phylogenetic correlation matrix that was derived from the phylogenetic tree[81].

We constructed the phylogeny across all vertebrate classes present in our data by using the *rtrees* R package[82] to randomly draw one phylogeny from the megatrees available for each phylogenetic class. We used the following megatrees: for birds: Jetz et al.[83], for mammals: Upham et al.[84], for fish: Rabosky et al.[85] and for reptiles: Tonini et al.[86]. Since for reptiles *rtrees* only has the phylogeny for squamates, we added the turtle phylogeny from Thomson et al.[87]. Once we obtained a randomly drawn mega-tree per class, we then grafted them on the order-based backbone for which the node times were taken from the Timetree of life (https://timetree.org/).

To avoid overfitting, we refitted meta-analytical models without accounting for phylogeny. We then compared both models and retained the one that fitted the data best based on the lower marginal

AIC. To account for uncertainty in available megatrees (e.g., birds, mammals) that were used to build the synthetic phylogeny, we randomly selected 100 trees from available posterior phylogenies[82] and thus fitted both types of models 100 times. For the models that accounted for phylogenetic relatedness we calculated Pagel's $\lambda$ following Cinar et al.[81], as $\lambda = \frac{\sigma_p^2}{(\sigma_n^2 + \sigma_p^2)}$; where $\sigma_p^2$ is between-species variance due to the phylogeny and $\sigma_n^2$ is between-species variance due to other effects, e.g. the environments the species inhabit.

We accounted for uncertainty in the path coefficients estimated with SEMs by weighting the residual variation by the inverse of the squared SEs of the path coefficients. Additionally, meta-analytical models with path coefficients that included climate as a dependent variable (i.e., those with the response variable CZ, CZG, CG and TotalCG) included a quantitative variable reflecting the probability of the climate signal being likely spurious ($P_{\Delta AICc}$) and a qualitative variable for climate data quality (two levels: 'exact' and 'approximate'). These meta-analytical models assess whether the intercept differs from 0, that is, whether each relationship in our path diagram differs from 0 across the studies. To test our expectation that phenological responses to temperature allow avoiding population declines, we assessed whether the proportion of studies with non-negative CZG was higher than by chance only. For this, we used a one-sided binomial test and tested whether the proportion of studies with non-negative CZG was greater than 0.5 for a given combination of climate variable and trait category.

All meta-analyses were fitted with the R package *metafor*[88], using the "uobyqa" optimiser. We tested significance with the Wald test. For each fitted model, we visually assessed the normality of residuals. We assessed the amount of heterogeneity among studies in our meta-analyses with commonly used metrics[31]: Higgins $I^2$ and the total amount of heterogeneity ($Q$), significance of which was estimated. Higgins $I^2$ reflects the proportion of total heterogeneity due to between-study variation. It ranges from 0 (heterogeneity is due to within-study variation only) to 1 (heterogeneity is due to between-study variation only) and is comparable among different meta-analyses. We re-run meta-analyses also separately for birds ($n = 84$) and mammals ($n = 9$). We could not run them for reptiles because of an insufficient sample size ($n = 4$).

### Explaining heterogeneity in CZ

We tested whether the high heterogeneity in the phenological sensitivity to temperature$_d$ (CZ) might be due to species responses differing with latitude or species-specific characteristics (such as migratory mode, diet, and generation time). Based on previous studies we formulated several a priori expectations. The expectations about migratory mode, generation time and latitude are detailed in the Introduction. Additionally, diet may also moderate phenological responses to temperature, with phenological shifts in herbivores previously shown to track the climate better than those of organisms at higher trophic levels[1,3].

To test our hypotheses about possible drivers of the heterogeneity in CZ, we fitted extended versions of mixed-effects meta-analytical models that were used to infer across-study effect sizes (see "Across-study inferences"). Specifically, we used the same random effects as in the mixed-effects models used to infer across-study effects (see "Across-study inferences") and propagated uncertainty in the same way. We additionally included as predictors the variables hypothesised to affect the phenological responses to temperature, i.e. absolute latitude (quantitative, in degrees), generation time (quantitative, in years), diet (qualitative, three levels: herbivore, carnivore, omnivore), and migratory mode (qualitative, two levels: resident and migrant; for data sources see Supplementary Note 7: Sources of species-specific characteristics). As the response variable we used study-specific CZ. The model also included as a covariate the

quantitative variable reflecting the probability of the climate signal being spurious ($P_{\Delta AICc}$). We grand-mean centred (i.e. subtracted the overall mean from each value) the quantitative predictors prior to fitting the model, so that the intercept estimate reflects the mean value of the CZ path and slopes reflect the change in traits per change in one original unit of the predictors.

We expected that species characteristics and latitude explain not only sensitivities of phenology to temperature (CZ) but also other paths in the path diagram, and, particularly, trait-mediated effects of climate on population growth rate (CZG). Therefore, analogous models were also fitted with ZG, CZG and CG used as a response variable. The statistical inference and model diagnostics were performed in the same way as for the meta-analytical models with intercept only. The overview of all meta-analytical models fitted in this study is given in Supplementary Table S8.

### Reporting summary
Further information on research design is available in the Nature Portfolio Reporting Summary linked to this article.

### Data availability
The sTraitChange dataset generated in this study has been deposited on Zenodo as part of the R project 'sTraitChange_Analyses' under https://doi.org/10.5281/zenodo.17629266[89]. See the Readme of the R project 'sTraitChange_Analyses' for more information. The overview of the dataset is provided as Supplementary Data 1.

### Code availability
The functions required for the analyses and the complete workflow needed for this study are made publicly available, respectively, as part of the R package 'sTraitChange'[90] and R project 'sTraitChange_Analyses'[89].

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

## Acknowledgements

This manuscript is a result of the sDiv-funded group sTraitChange and two workshops supported by sDiv (Synthesis Centre at German Center for Integrative Biodiversity Research (iDiv), Jena-Halle-Leipzig). We are grateful to the many researchers who collected field data and kindly shared them. Among others, we would like to thank Rocky Rockwell, Gregory Brown, I-Jiunn Cheng, Wolf, Vladimir Grosbois, Eric Hansen, Oliver Kruger, Andrew Cockburn, Scott Sillett, Heather Renner, Don Dragoo, William J. Sydeman, Cristina Rodríguez Juarez and the sTraitChange Data Consortium (a full list of consortium members appears in Supplementary Note 1). Without such contributions, this study would have been impossible. Sadly, M.H. and D.v.H. passed away during the development of this manuscript. Their long-term datasets are included in this study, and we are indebted to their invaluable contribution. We thank Thomas Banitz for his participation in the sTraitChange workshops and contributions to this study. We thank the UKCEH National Capability for UK Challenges programme NE/Y006208/1 and the Joint Nature Conservation Committee for financial support. This study is part of the long-term Studies in Ecology and Evolution (SEE-Life) program of the CNRS. We acknowledge funding from NSF ORCC project # 2222057 to S.J., NSF DEB project 0089473 to F.S.D., Research Council of Finland grant SA338180 to T.E., Swedish Research Council grant 2021-03892 to J.H.-S., ERC Starting Grant 639192-ALH to T.E.R., Research Council of Norway (project 223257) and the European Research Council (ERC-2022-AdG-101095997) to B.-E.S., Natural Environment Research Council, UK to J.P., Funding by Universidad Nacional Autónoma de México (PAPIIT IN211491, IN-200702-3, IN206610-3, IN205313 and IN205819), Consejo Nacional de Ciencia y Tecnología (81823, 47599, 34500-V, 4722-N9407, 104313) and National Geographic Society (991416) to H.D., Natural Environment Research Council, UK to L.K., BOF-Methusalem grant (48098) to E.M., UID/04292—Centro de Ciências do Mar e do Ambiente (MARE), LA/P/0069/2020 to the Associate Laboratory ARNET, and from the Falkland Islands Government

 

to P.C., NSF DEB-1242510 to F.J., the Spanish Ministry of Science and EU FEDER funds (PID2021-122893NB-C21) to D.O., Spanish Ministerio de Ciencia e Innovación/AEI and EU-FEDER funds (PID2021-124731NB-I00, PIE2022301133) to M.G., and Fundação para a Ciência e a Tecnologia I.P. via cE3c (DOI: 10.54499/UIDB/00329/2020) and CHANGE (LA/P/0121/2020), national funds, and the co-funding by the FEDER, within the PT2020 Partnership Agreement and Compete 2020 to J.P.G. S.R.B. and K.S.B. acknowledge multiple grants from the National Science Foundation, Smithsonian Institution, and National Geographic Society, as well as support from the Maxwell-Hanrahan Foundation for parrotlet studies. Data from Pointe Géologie were collected with the logistical and financial support from Institut Polaire Français Paul-Emile Victor (IPEV), Terres Australes et Antarctiques Françaises, and Zone Atelier Antarctique et Terres Australes (LTSER France). Data from Montpellier was collected with long-term support from the OSU-OREME.

## Author contributions

The sTraitChange workshop participants (V.R., C.V.J., N.Mc.L., A.Cha., C.T., L.B., U.B., G.C., K.K., S.K.S., E.M., S.Or., H.S., S.J.G.V., B.-E.S., S.J., J.C., S.R.B., M.E.V., T.E.R., and M.v.d.P.) have designed the study, V.R., C.V.J., G.C., E.M., T.E.R., J.C., S.K.S., and N.Mc.L. have conducted systematic literature review, C.V.J. with help of V.R. have extracted the data from the papers and collated the dataset, V.R. with help of T.E.R. and M.v.d.P. have analysed the data, with feedback from A.Co. A.Cha., C.T., R.A., S.A., T.A.-N., P.A., D.A., L.M.A., C.B., K.S.B., D.B., D.T.B., S.B., L.B., P.C., A.Chi., F.D., K.D., F.S.D., H.D., T.E., D.F., G.G., M.G., J.P.G., J.H.-S., M.H., J.M.I., F.J., E.K., L.K., S.L., M.Mal., J.M., M.Mas., E.M., J.-B.M., A.P.M., C.R.N., M.N., S.Op., D.O., D.P., T.P., A.P.-P., J.P., R.A.P., N.P., J.M.A., H.G.R., A.S.-A., C.Sa., C.Sch., J.S., B.C.S., G.T., C.E.T., V.V., V.A.V., D.v.H., S.J.G.V., S.W., N.W., A.G.W., B.-E.S., S.J., J.C., S.R.B., and M.E.V. contributed their data to the study. The sTraitChange workshop participants have discussed intermediate results and contributed to their interpretation. V.R. has produced the first manuscript draft, and all co-authors contributed to revisions.

## Funding

## Competing interests

The authors declare no competing interests.

## Additional information

Viktoriia Radchuk [1] ✉, Carys V. Jones[2], Nina McLean [3], Anne Charmantier [4], Céline Teplitsky[4], Ray Alisauskas [5], Sergio Ancona [6], Tycho Anker-Nilssen[7], Peter Arcese [8], Debora Arlt [9], Lise M. Aubry [10,11], Liam Bailey[1], Christophe Barbraud[12], Karl S. Berg [13], Dominique Berteaux [14], Daniel T. Blumstein [15,16], Sandra Bouwhuis [17], Ulrich Brose [18,19], Lyanne Brouwer [20], Paulo Catry [21], Guillaume Chero[1], Andre Chiaradia [22,23], Alexandre Courtiol [1], Francis Daunt [24], Karine Delord[12], F. Stephen Dobson [25], Hugh Drummond[6], Tapio Eeva [26], Dominique Fauteux [27], Gilles Gauthier [28], Meritxell Genovart [29], José P. Granadeiro[30], Jonas Hentati-Sundberg [31], Michael Harris[24,64], José Manuel Igual [32], Fredric Janzen [33], Katharine Keogan[34], Erkki Korpimäki[26], Stephanie Kramer-Schadt [1,35], Loeske E. B. Kruuk[34], Sue Lewis [24,34,63], Mark Mallory [36], Julien Martin [37], Manuel Massot [38], Erik Matthysen [39], Jean-Baptiste Mihoub [40], Anders Pape Møller [41], Chloé R. Nater [42], Mark Newell [24], Steffen Oppel [43,44], Daniel Oro [29], Santiago Ortega [6], Deseada Parejo [45,46], Tomas Pärt [9], Ana Payo-Payo[47], Josephine Pemberton [34], Richard A. Phillips [48], Neville Pillay [49], Jesús M. Avilés [45,46], Heiko G. Rödel [50,51], Ana Sanz-Aguilar [32,52], Claire Saraux[53], Holger Schielzeth [19], Carsten Schradin[49,53], Julia Schroeder [54], Ben C. Sheldon [2], Giacomo Tavecchia [32], Corey E. Tarwater [55], Vebjørn Veiberg [7], Vincent A. Viblanc [53], Dietrich von Holst[51,64], Stefan J. G. Vriend [56], Sarah Wanless[24], Nathaniel Wheelwright[57], Andrew G. Wood[48], Bernt-Erik Sæther[58], Stephanie Jenouvrier [59], Jean Clobert[60], Steven R. Beissinger [61], Marcel E. Visser [56], Thomas E. Reed [62,65] & Martijn van de Pol [20,65]

[1]Leibniz Institute for Zoo and Wildlife Research (IZW), Berlin, Germany. [2]Edward Grey Institute, Department of Biology, University of Oxford, Oxford, UK. [3]Office of Nature Conservation, Environment Planning and Sustainable Development Directorate, ACT Government, Canberra, ACT, Australia. [4]CEFE, Univ Montpellier, CNRS, EPHE, IRD, Montpellier, France. [5]Environment and Climate Change Canada, Prairie and Northern Wildlife Research Centre, Saskatoon, SK, Canada. [6]Instituto de Ecología, Universidad Nacional Autónoma de México, Mexico City, Mexico. [7]Norwegian Institute for Nature Research,

Trondheim, Norway. [8]Department of Forest and Conservation Sciences, The University of British Columbia, Vancouver, BC, Canada. [9]Department of Ecology, Swedish University of Agricultural Sciences, Uppsala, Sweden. [10]Department of Fish, Wildlife and Conservation Biology, Colorado State University, Fort Collins, CO, USA. [11]Leverhulme Visiting Professor, Centre for Ecology and Conservation, University of Exeter, Penryn Campus, Penryn, Cornwall, UK. [12]Centre d'Etudes Biologiques de Chizé, UMR 7372 CNRS-La Rochelle Université, Villiers-en-Bois, France. [13]School of Integrative Biological and Chemical Sciences, University of Texas Rio Grande Valley, Brownsville, TX, USA. [14]Centre for Northern Studies, Université du Québec à Rimouski, Rimouski, QC, Canada. [15]Department of Ecology and Evolutionary Biology, University of California, Los Angeles, CA, USA. [16]The Rocky Mountain Biological Laboratory, Crested Butte, CO, USA. [17]Institute of Avian Research, Wilhelmshaven, Germany. [18]EcoNetLab, German Centre for Integrative Biodiversity Research (iDiv) Halle-Jena-Leipzig, Leipzig, Germany. [19]Institute of Biodiversity, Ecology and Evolution , Friedrich Schiller University Jena, Jena, Germany. [20]College of Science and Engineering, James Cook University, Townsville, QLD, Australia. [21]MARE—Marine and Environmental Sciences Centre/ARNET—Aquatic Research Network, Ispa—Instituto Universitario, Lisboa, Portugal. [22]Conservation Department, Phillip Island Nature Parks, Cowes, VIC, Australia. [23]School of Biological Sciences, Monash University, Clayton, VIC, Australia. [24]UK Centre for Ecology & Hydrology, Bush Estate, Penicuik, UK. [25]Department of Biological Sciences, Auburn University, Auburn, AL, USA. [26]Department of Biology, University of Turku, Turku, Finland. [27]Arctic Centre, Canadian Museum of Nature, Gatineau, QC, Canada. [28]Centre for Northern Studies and Department of Biology, Université Laval, Québec, QC, Canada. [29]Department of Ecology and Complexity, Centre d'Estudis Avançats de Blanes (CEAB - CSIC), Blanes, Girona, Spain. [30]Centre for Ecology, Evolution and Environmental Changes, Departamento de Biologia, Faculdade de Ciências, Universidade de Lisboa, Lisboa, Portugal. [31]Department of Aquatic Resources, Swedish University of Agricultural Sciences, Uppsala, Sweden. [32]Animal Demography and Ecology Unit, IMEDEA (CSIC-UIB), Esporles, Spain. [33]Departments of Fisheries and Wildlife & Integrative Biology, Kellogg Biological Station, Michigan State University, Hickory Corners, MI, USA. [34]Institute of Ecology and Evolution, University of Edinburgh, Edinburgh, UK. [35]Institute of Ecology, Technische Universitaet, Berlin, Germany. [36]Acadia University, Wolfville, NS, Canada. [37]Department of Biology, University of Ottawa, OttawaON, Canada. [38]Institut d'Ecologie et des Sciences de l'Environnement de Paris, Sorbonne Université, CNRS, INRAe, Paris, France. [39]Evolutionary Ecology Group, University of Antwerp, Wilrijk, Belgium. [40]Centre d'Ecologie et des Sciences de la Conservation (CESCO), Sorbonne Université, MNHN, CNRS, Paris, France. [41]Ecologie Systématique Evolution, Université Paris-Sud, CNRS, AgroParisTech, Université Paris-Saclay, Orsay, Cedex, France. [42]Department for Terrestrial Biodiversity, Norwegian Institute for Nature Research (NINA), Trondheim, Norway. [43]Royal Society for the Protection of Birds, Centre for Conservation Science, David Attenborough Building, Cambridge, UK. [44]Swiss Ornithological Institute, Sempach, Switzerland. [45]Department of Functional and Evolutionary Ecology, Experimental Station of Arid Zones (EEZA-CSIC), Almería, Spain. [46]Unidad Asociada (CSIC-UNEX): Ecología en el Antropoceno, Badajoz, Spain. [47]Departamento de Biodiversidad, Ecología y Evolución, Facultad de Ciencias Biológicas, Universidad Complutense, Madrid, Spain. [48]British Antarctic Survey, Natural Environment Research Council, High Cross, Cambridge, UK. [49]School of Animal, Plant and Environmental Sciences, University of the Witwatersrand, Johannesburg, South Africa. [50]Laboratoire d'Ethologie Expérimentale et Comparée UR 4443 (LEEC), Université Sorbonne Paris Nord, Villetaneuse, France. [51]Department of Animal Physiology, University of Bayreuth, Bayreuth, Germany. [52]Department of Biology, University of Balearic Islands, Palma, Spain. [53]Université de Strasbourg, CNRS, IPHC UMR 7178, Strasbourg, France. [54]Department of Life Sciences, Imperial College London, Silwood Park Campus, Ascot, UK. [55]Department of Zoology and Physiology, University of Wyoming, Laramie, WY, USA. [56]Netherlands Institute of Ecology (NIOO-KNAW), Wageningen, The Netherlands. [57]Department of Biology, Bowdoin College, Brunswick, ME, USA. [58]Gjærevoll Center for Biodiversity Foresight Analyses, Norwegian University for Science and Technology (NTNU), Trondheim, Norway. [59]Biology Department, Woods Hole Oceanographic Institution, Woods Hole, MA, USA. [60]Station of Experimental and Theoretical Ecology (SETE), UMR 5321, CNRS and University Paul Sabatier, Moulis, France. [61]Department of Environmental Science, Policy and Management, 130 Mulford Hall, University of California, Berkeley, CA, USA. [62]School of Biological, Earth and Environmental Sciences, University College Cork, Cork, Ireland. [63]Present address: Centre for Conservation and Restoration Science, School of Applied Sciences, Edinburgh Napier University, Edinburgh, UK. [64]Deceased: Michael Harris, Dietrich von Holst. [65]These authors contributed equally: Thomas E. Reed, Martijn van de Pol. ✉e-mail: radchuk@izw-berlin.de

