## [Transparent Peer Review file · Nature Communications]

Changes in phenology mediate vertebrate population responses to temperature globally

Corresponding Author: Dr Viktoriia Radchuk

Version 0:

Reviewer comments:

Reviewer #1

(Remarks to the Author)

The authors do a great job addressing a timely question, by pulling together an impressively broad dataset and using a sophisticated multi-step analysis to arrive at a meta-analysis that teases apart direct and indirect effects of climate on vertebrate population growth rates. I am impressed that the authors were able to find so many species with concurrent population size and trait time series; I was only aware of a handful of these previously. The work should be of interest to a broad audience.

I think the simple SEM (conceptually simple, not so simple to implement) is the correct way to approach these data, and I am impressed by the care the authors took to propagate estimate uncertainties into the meta-analysis and the range of covariates that were used in the meta-analysis. I have no concerns about the analyses, and the script files are well annotated. I'm not at all surprised that the phylogeny didn't improve model fit, but it was worth testing anyway, and I applaud the extra work that went into its inclusion. I also appreciate the inclusion of the null models and intercept-only models as reference points.

Between lines 439 and 440, or somewhere else appropriate, please add the summary information from Fig. S16, that the median window duration was 2 or 3 weeks, for temp and precip respectively. I think that result belongs in the main text.

At L407, please add the range of grid sizes to the main text (0.05 to 0.5 degrees), so that readers don't have to check Table S6.

The manuscript is written well. Here are a couple of minor suggestions for awkward places I spotted, but I had no major concerns about the writing.

L519: I'd use 'among studies' instead of 'between studies' (as you did on L600).

L551: should be 'probability ... was' or 'probabilities ... were'

The order of panels in Fig. 3 is odd. Wouldn't it be just as clear to have panel a) in the upper left, and swap it with panel b)? I don't think the arrows showing relationship to axes are necessary. In panel a) it's suspicious that none of the phenologies are insensitive to climate, i.e. the blank gap around a slope of 0; the same gap is present in panel c) of course. Is that gap a sign of publication bias? Other meta-analyses of phenological change don't show that gap. I'm also curious whether there is any phylogenetic signal to the variation in the slopes of panel a). Could those individual lines be color coded instead of all grey, to represent the groups of vertebrates (birds, mammals, fish)? A recent paper suggested there may be some phylogenetic signal there (Loughnan, D., S. Joly, G. Legault, H. M. Kharouba, M. Betancourt and E. M. Wolkovich. 2024. Phenology varies with phylogeny but not by trophic level with climate change. *Nature Ecology & Evolution* 8(10): 1889-1896.).

In Fig. S8 some of the models have extremely high R² values. Is this potentially because in Fig. S9 many of the species have very strong density dependence (PG effects are much larger than any others)? While density dependence is biologically important, it would be nice to know how much of the high R²s is due to the PG arrow rather than CZG or CG.

In Fig. S9 it would be nice to see which effect sizes for the different arrows belong to which species. I'm wondering if lines

that connected the points within a column would reveal patterns of covariances among those, for example species with higher CZ would also have higher ZG, or vice versa? Of course, its also possible that that many lines would be a horrible visual mess and not lend any insights, but it may be worth checking. If there are any strong covariances among those (which would be easy to check), that would be worth mentioning in the main text too.

In S16, can the length of the window be related to generation time, or something else about the pace of life? I'm surprised by the length of the longer windows.

(Remarks on code availability)

The data are accessible, and the scripts look to be well annotated. There are MANY script files, and I did not open all of them. The ones I checked looked good.

Reviewer #2

(Remarks to the Author)

This paper uses path analyses and meta-analytical approaches to explore phenotypic trait mediated climate impacts on population growth in vertebrates. They used 213 time series of phenological and morphological traits to test the path of influence of temperature and precipitation effects on population growth. They found highly variable results across the species and locations. In general, phenological events were more climate influenced than morphological traits, which showed no clear trait-mediated climate effects. The most noteworthy results are that phenology was predominantly temperature driven with then positive impacts on population growth, suggesting adaptative phenological responses to temperature change in many species. This effect of phenology was however shown to be weaker at lower latitudes. Additionally, morphology was insensitive to climate changes.

Here, I begin with some overarching summaries covering topics requested in the reviewer guidelines, before giving a detailed review.

This study provides some original insights into the topic of climate change impacts on animal populations growth, thus giving an important contribution into the field. It is a novel study researching trait-mediated effects of temperature at a global scale and for multiple populations. The findings from this work will be of significant value in the field and related fields in demonstrating why researchers need to consider different pathways of influence of climate effects. It is also a pertinent reminder to consider population level impacts. The methodology is sound, and I would say exceeds expected standards in the field.

The paper is well written, easy to follow, and has a good flow. The methodology in particular explains some very complex and detailed analyses in a clear and concise way, and I would like to complement the authors on this. The data analyses used in this paper are really clearly written and explained. These are complex analyses but well thought through and supported with a well-documented R package, which is to be commended! I would especially like to highlight the treatment and propagation of uncertainties as a really nice part of the method. However, the discussion of the previous literature to support the context and claims (primarily in the introduction and discussion) could do with improvement. Specifically, the taxonomic scope of the references is quite limited, particularly to bird species, and given the broad scope of the paper, this should be reflected in the cited literature. Examples of particular areas where this could happen are detailed below but I would recommend reviewing the whole introduction and discussion with this in mind as well. Following from this, the work conducted does support the broad conclusions, however, there is some over-generalisation of some of the conclusions, e.g. not take into full account the limitations of the (granted very good) dataset e.g. geographically and taxonomically biased. Please also see other comments about using averages to draw conclusions from highly variable inputs and results.

There are a few areas where I believe this manuscript would benefit from addressing, which I detail below. Any consideration of the points mentioned would require minor revision and should not prohibit publication.

Main areas for improvement:

1. Expanding literature support, particularly increasing taxonomic scope of cited literature. There are also some cases where claims made are not currently sufficiently supported.

The authors could, for example, pitch their study importance more effectively in the introduction, by highlighting how the importance of looking at population growth effects has been previously underscored in the literature, yet there is a need for more studies to test this (e.g., Samplonius et al., 2021). Samplonius, J.M., Atkinson, A., Hassall, C., Keogan, K., Thackeray, S.J., Assmann, J.J., Burgess, M.D., Johansson, J., Macphie, K.H., Pearce-Higgins, J.W. and Simmonds, E.G., 2021. Strengthening the evidence base for temperature-mediated phenological asynchrony and its impacts. *Nature Ecology & Evolution*, 5(2), pp.155-164.

L295 – This is a good example of lack of support, reference 34 is one example of one species and one phenological trait that is being used to support a statement that phenological traits in general are plastic. Either change the claim or add greater

support from a wider range of taxa and phenological events. (There are many other examples like this and I will not have found them all).

L312 – again a single reference of support to a very general claim.

The support for the use of morphological traits and particularly those chosen, several of which seem that they would not change within life beyond developmental changes e.g. snout-vent length, is not sufficient.

e.g. L184 – not many references chosen and not linked to morphological traits included. L186 – could not see any supporting references for morphological traits beyond body mass and body condition (which is actually calibrated using body length, further suggesting this trait should not be considered to be labile or plastic). Support for snout-vent length, body length, and bill length (another which I'm not sure should change interannually in response to weather conditions) was not given.

2. Consider whether the averaging of results in the meta-analysis is a meaningful metric to assess the results.

Averaging the path coefficients to give (what I assume is a mean) result, I feel, is not the most meaningful metric. This is particularly the case given the different phenological or morphological traits being considered and due to the variability of results where a mean of 0 could result from all results being close to 0 or from as many above 0 as below. By saying that the average of 0 indicated adaptive responses does not give more insight than looking at the number of non-negative results but it does perhaps have some biological issues. The traits being considered are quite different, it is perhaps not to be expected that things like lay date would have the same response and effect as emergence date or parturition, in the same way body mass would not be expected to respond in the same way as body length or bill size. Therefore, comparisons among these responses or analysis of modal or majority trends are ok but direct averaging ignores these differences.

L233-236 – this is an example where I do not find the averaging so helpful as it you may not expect all traits to behave the same, and a lack of an average response in a particular direction does not mean that for individual species and traits that they are not adaptive.

L295-297 – again, not sure about the averaging particularly given the highly variable results. I would find it more meaningful to look at majorities of directions or between group differences etc. The average does not really say anything biologically.

3. Reduce over generalisations. E.g. saying phenology when mean spring phenology.

This study only looks at spring phenological events (as far as I can tell) but this is not mentioned anywhere. This needs to be mentioned from the very beginning as it strongly impacts the interpretation of the results. It is not a drawback, just a clarification that is important.

L141 – this is certainly one example of the impact of phenology; however, it is very specific and it does not hold for all biological events e.g. those in Autumn or if caterpillars hatch prior to their food source. Please be more specific about the scope of events you cover – just spring phenology in a few species. The caterpillar example is still spring but would not always fit this pattern.

L289 – this is a bit over generalised due to the limitations mentioned below, in particular geographic and taxonomic biases. In addition, it is only a subset of spring phenology that has been looked at. I suggest moderating the language here to focus on spring phenology, clarify that vertebrates are mostly birds, and across most of the globe (not all). Given that you have results for individual studies – can you say for which groups these temperature mediated effects are most clear?

L596-603 – I find this a more specific result and interesting, I think this should be elevated to the main text not hidden in the methods.

4. Include more comment or a specific section on the study limitations. e.g. geographic spread, taxon spread, type of morphological traits chosen – possible not plastic or labile (some of the references they have use body length to standardise body mass to look at body condition). Also include the different types of phenology and morphology and whether they can truly be compared.

The geographic limitations of the dataset need to be acknowledged early, especially the lack of studies in the tropics and from low latitudes and the bias to Europe and North America. This is particularly important given the latitude effect that was found, some comment on whether this effect would hold into the tropics or with more data would be good. E.g. L283 it mentions close to the equator, but it seems there is limited data in this study from the equator.

Another potential limitation of the study is the substantial variation in climate data resolution across the countries considered. Specifically, the resolutions for North America, South America, and Africa are relatively coarse. Such large-scale temperature data may not accurately capture the environmental conditions experienced by animal populations, particularly the microclimates to which they respond. Please include a small mention of this.

L318 – this comes in very late and is not fully explained. What does this mean for all other results?

L399 – 402 – I feel that these limitations need to be mentioned further up somewhere.

Minor comments:

General

L130 – I would replace ‘global warming’ with ‘climate change’, I appreciate it is said a few times in these sentences but it is a more accurate characterisation of the actual changes that will be experienced.

L311 – the population is not plastic or not, the trait is.

L312-315 – I do not agree with the reasoning here. Surely physiology mediates or plays a role in all traits, particularly phenological traits and therefore, it will be playing a role at all latitudes. I do not see physiology and phenology as separate, physiological processes will be the basis of the phenological events we observe (e.g., gonadal development for laying date). Perhaps you could be more specific about which physiological processes you think are influencing things at lower latitudes. Or perhaps the authors are referring to physiological effects as direct temperature effects and phenological effects as indirect temperature effects (e.g., mediated through food supply). If so, then perhaps this distinction could be clearer.

L344 – again, not sure about this distinction, could you be specific about the physiological traits. Also again, phenological traits will be built up from physiological traits and indeed in some species they could be the same e.g. trees and invertebrates. Perhaps add a qualifier about vertebrates here.

L252 – what about empirical models? I do not think this is a clear distinction as models can be applied to empirical datasets, did you mean theoretical or simulation studies?

L380 – this link does not work – takes me to a 404 page

Data and methodology

More detail on how the biological realism of the identified climate windows would be a nice addition. While searches like sliding window can identify climate drivers that make excellent explanatory and predictive models, they are not always biological realistic. In this paper some choices around the window searching could lead to biological realism being reduced, for example, 2-year search could actually exceed the lifespan of some short-lived species in the dataset, or at least be before they are born when they reach maturity at age 1. In addition, the use of a reference day that is the latest date a phenological event was observed will in many years be occurring after the event takes place. Some discussion of the biological realism and implications for interpretation would be a good addition.

Why was the detrending of the climate variables done after the sliding window analysis not before? I am curious as the given rationale would seem to hold for detrending prior to the sliding window.

L257 – p-values are not typically indicative of trend directions alone (could be in a one tailed test of some kind). Is this what is meant here, or did you mean beta value? If you did mean p-value, maybe explain how it indicates the trend direction.

L406 – please mention that the resolutions vary here.

L438 – please provide these preliminary analyses somewhere like SI.

L537 – is the difference to 0 something you would expect?

L540 – could you give a bit more detail on the binomial test please e.g. I assume it has an assumed probability of some kind?

L579 - the same way as what? The reference here is not clear.

Conclusions

See above on generalisations.

Figures

Figure 1 – Conceptual diagrams need to be really clear. The overlapping of arrow heads with boxes makes this less so, could you add just a tiny bit of breathing room around each arrow head please. Esp in panel a under the top box.

Figure 3 – The order of the panels here is not very clear. Perhaps the letters could be changed to match the order – that makes it easier for the reader to assimilate the information. This figure also needs tidying, the arrows are different sizes and the panels are not full aligned.

Supporting information

Supplementary figure S9: and any similar figures. I find the colour and shape differentiation very hard to see. Please choose some that are more distinct. At the size they are, even on a big monitor, I cannot easily see the difference between the circle

and the square, nor light-dark blue/red.

L313 – I found this conceptual framework hard to follow and it did not seem to fit with the results in the main text. I hope I did not miss something, if I did, please correct me. But especially given that the climate data were detrended in the main text, I do not see how these two scenarios were tested. Scenario is also not mentioned in the main text. I suggest removing this section.

L340 – GR is not defined.

(Remarks on code availability)

I have looked through the repository but have not tried to run all of the code. The repository is well organised with a nice README to begin and all code looks great. The functions I have looked at are really comprehensively commented and clear.

Reviewer #3

(Remarks to the Author)

(Remarks on code availability)

Reviewer #4

(Remarks to the Author)

In this manuscript, the authors Radchuk et al. conduct a meta-analysis using time series of vertebrate phenology, morphology, and population growth. Using this diverse data, the authors test the relationships between temporal and morphological traits, climate, and population dynamics. Morphological traits showed no relationship to temperature or precipitation and population trends, but a relationship was found between temperature and phenology---but not phenology and population responses. These results suggest that other traits are shaping population responses to climate change than those represented in the current literature.

Understanding the relationships between traits and population trends is important, as it may provide insights needed to forecast and manage species under climate change. Synthesizing trends globally and across diverse species assemblages also allow us to identify general trends and large-scale spatial patterns. The dataset presented in this study is impressive, particularly given the efforts to collect raw data from the original authors. However, I was surprised that the phylogenetic signal was weak. There is strong evidence of phylogenetic conservatism in phenology, with the variation due to latent traits potentially reflecting traits favourable under ancestral climates. While I appreciate you did include a phylogeny in your analysis, there was no justification or discussion of the importance of evolutionary history in the introduction. I would have like more detail on exactly how phylogeny was included in the model, whether specific packages were used, and what the estimated phylogenetic structure was. As an additional supporting analysis, I am curious whether relationships would be stronger within the individual taxonomic groups, especially given the diversity in the types of phenological events included in the study.

The framing of the project and writing could also be improved in my opinion. I felt the state of the literature could be better represented and some of the broad statements tempered. I disagree that no work has been conducted on this topic before. But perhaps more elaboration and nuance would address this, as I do believe that there are few robust and global studies on this topic and appreciate the novelty of these large-scale studies. The discussion of the results would also benefit from more nuance, as the lack of statistical significance was at times not clear from how trends were discussed. Finally, the discussion would benefit from more discussion of future directions and applications of these findings to conservation or species management.

Specific comments:

The title should more accurately reflect the study results. The current title makes it seem like the relationships between temperature and precipitation were strong, but in reading the text, it was only temperature.

Line 110: More context for your study is needed here, it sounds like it was exploratory, but I am sure a lot of thought when into why you collected all these time series

Line 114-119: I think these sentences could be edited to better guide your reader and make it clearer that you are testing several relationships and why.

Line 127: This introductory paragraph introduces some big topics that are not fully developed. Could you break this into multiple paragraphs and further develop some of these topics?

Line 132-135: This should be its own paragraph to convince the reader this is important and true.

Line 141: Earlier phenologies can have huge costs as well, your argument would be stronger if you include some of these examples.

Line 145: What is meant by “net demographic consequences of interannual variation in phenotypes”? Is there not an existing literature of traits being included in integral projection models to do exactly this?

Line 150-152: Could you elaborate and give an example of this? How common is this actually?

Line 165: I don't agree that it has “not been investigated” and would consider even some of your own coauthor's work (e.g. Jenouvrier et al. 2018. Journal of Animal Ecology) to have done so. I would temper sentences like this and elaborate on them to more accurately reflect the state of the field or provide more nuance to your position.

Line 178: Can you elaborate on why?

Results:

Line 196: The use of the term “study” is misleading, would it not be more appropriate to refer to 116 time-series for example?

Line 204-236: This text would be better integrated into the introduction. The text also does not reference any of the other panels in Fig 1 and readers would benefit from more detail. Or for the results to be discussed in the context of this figure.

Line 244: Do you mean “found for individual studies” or interannually for a given study?

Methods:

Line 394: What are the four classes? More detail, especially sample sizes, would be useful.

Line 399-402: Could you add the percentage of each taxonomic group and types of phenological events?

Discussion:

Line 290-291: This is a key point that is missing from the introduction.

Line 303: What are the units?

Line 520: How was phylogenetic relatedness estimated? Was a specific package used? What was the estimated value of lambda?

Line 564-575: This provides important context and justification that might be better suited in the introduction.

Line 256, line 440: Minor typos

Figures & Tables:

Fig 1: Does the slope and the direction of the trends in b-d reflect your predictions?

Does the order of the subplots, which I initially read as c, a, d, b, have any significance. I think figure a would be clearer if it was separated from b-d into different panels.

Figure 2:

The inset should be wider to better separate the two categories. As is, the three words read more like the x-axis label.

This figure is very visually appealing, but the font color makes some of the diet letters hard to read. I also did not feel the black bars were necessary as the numbers themselves are present and easily interpretable.

What does the diet “LC” stand for?

Figure 3:

To better illustrate the relationship between a and c, I would put panel a under panel c and have a upward pointing arrow.

Figure 4:

Line 828-831: As written the caption makes it sound like there is a trend in panel a, but it is also not significant. This could be more clearly written to clearly reflect the statistics.

Having the full terms on the axes, not the letter acronym, would make these plots easier to understand with a quick glance.

(Remarks on code availability)

I looked through the GitHub repository and several of the R code files. The repo structure is clear and well organized. It also includes a README that identifies all the key files.

Looking through the code, I thought it was well formatted and would be easy to run if I had the climate data.

Version 1:

Reviewer comments:

Reviewer #1

(Remarks to the Author)

The authors did an excellent job responding the review comments, and I have no concerns about these revisions. The text and figures are both improved, such that main points and results are clearer now. I think this work contains multiple important messages for our field, and I hope that it is widely read.

(Remarks on code availability)

Again, the extensive and well-annotated code should be a good resource for future studies.

Reviewer #2

(Remarks to the Author)

Response to response

All line numbers are for track changes doc

I would like to start by thanking the authors for their efforts in addressing my comments. There has clearly been a lot of work put in particularly to improve the diversity of literature cited, to reduce the over-generalisation, and to have a better acknowledgement of the study limitations. However, there are a few conceptual and interpretation concerns that I do not feel were sufficiently addressed. I'm afraid I also have an additional request/suggestion for the supporting information that has arisen with the addition of new components. Overall, I still think this will be an important paper, but this means it is even more crucial to make sure it is as robust as possible, and no conclusions are misleading.

The key areas that I feel were not yet addressed sufficiently were (split into changes I suggest and then rationale for changes):

Justification of morphological traits considered here (Comment 8).

Proposed changes:

Add more contextual discussion on the morphological traits considered to the introduction (like L179-201 for phenology). Should include some expectations and discussion of the different ways morphological traits could show interannual variation at the population level – as in response to comment 8.

Rationale: In the response to reviewer comments the justification for including non-labile traits put forward was population level mean changing between cohorts possibly due to labile or developmental plasticity. However, these suggested mechanisms are quite different processes to the other traits considered (phenology and labile morphology e.g. body mass). Therefore, greater discussion of the morphological traits is required in the manuscript itself. At present the introduction reads like a phenology introduction – a good one for phenology – but missing morphology. Morphology is only mentioned 3 times in the whole introduction, once to say morphology responds to climate change then to introduce the data and approach – I think this can leave a reader questioning the inclusion of morphology, particularly the variety of traits. As an example, the authors could expand on the call for more studies in the Teplitsky and Millien (2014) paper they cite to help support the inclusion of morphology.

Interpretation and conclusions drawn from meta-analysis average effect (Comment 9, 10, 11).

Proposed changes 1: remove or tone-down/carefully caveat the conclusion statements e.g. 'demonstrate that phenology mediates effects of temperature on population growth in vertebrates found across most of the globe', L367-369 (e.g. result of meta-analysis showed non-negative population growth response on average, but highly variable across studies with many showing negative response, partly driven by type of phenological event), L378-382 (I really don't think this part is needed – it just detracts from the other results – suggest remove), L472 (add in 'many' vertebrates).

Rationale: Thank you for your detailed responses here and clarifications on the meta-analysis. I think I can understand better your motivation and expectations for the meta-analysis. Here is my interpretation: you expected that you would be able to find a global effect size for trait-mediated responses to climate for phenology and morphology and that therefore the traits included and species etc were assumed to all be telling you about the same process – therefore a global effect made sense. I understand this motivation; but then we get to the results. At this point it is shown that there is actually a lot of variability in the process (probably even different processes going on) with some studies showing opposite patterns to others (for phenology and morphology). This is all very well assessed in the newly moved text in 'Explaining among-study heterogeneity in climate effects'. My issue is then that the first paragraph of the discussion where you conclude that you 'demonstrate that phenology mediates effects of temperature on population growth in vertebrates found across most of the globe'. I do not feel this a valid conclusion to draw. Yes, it is the result of the meta-analysis, but I think one has to consider

whether it is sensible to draw such a global or average result when there are influential structuring elements missing from the model e.g. phenological trait type, which you know are important. The conclusion could even be considered misleading making it sound like there is universal mediation, when in fact several studies show quite strong negative impacts on population size and misses all the nice nuance from the results section. I would recommend analysing the heterogeneity not reducing a global effect on something so variable, given the results found. If you wish to retain a global effect summary, I recommend it be clearly caveated as a statistical result rather than a biological conclusion. This is done well in the results already. I would like to add here that all the rest of the results from the path analysis and analysis of heterogeneity are already interesting and important enough, I do not feel collapsing to a global effect adds anything scientifically.

Proposed changes 2: Elevate the prominence of the meta-analysis as a key aim of the study (at end of intro), including discussion of how you think the 'global effect' can actually be interpreted biologically. Particularly in the case when effects of different sign are found between studies. But avoiding overgeneralised or misleading conclusions (where important and explainable variability is missed). Change terminology around 'average or on average' to better reflect the meta-analytical origin of the number e.g. global effect or meta-analysis effect or meta-analysis average.

Rationale: In the response the authors mention the importance of the average being a meta-analysis average. However, in the text, it is referred to it as 'on average' or just 'average' several times (e.g. L279, 314, 379, 1030). This explains my previous use of such language too. If the meta-analysis part is important, this should be elevated in the text so much misreadings are less likely. It should also be clearer in the text that a meta-analysis is a main aim of the paper. Currently that can be somewhat missed as the introduction positions the paper as if the main aim is a path analysis. Meta-analysis is not even mentioned there. This also happens in the results where the cross-study effect ZG is only mentioned in one sentence (and not made clear this is from the meta-analysis), but there is a large discussion of the binomial test results (paragraph at L307). This binomial test was what I referred to previously as the analysis of majority trends in comments 9 and 11, I was not suggesting vote counting. I had meant the results of the binomial test felt more informative than a global effect, but I now see the value of the heterogeneity results.

Proposed changes 3: following from Comment 10, when introducing the expectation, it would also be good to acknowledge that there are several ways to end up with a non-negative meta-analysis average, not just from all studies being adaptive to different degrees. It could arise from some highly adaptive results and others showing mild maladaptation, but still with a global effect that is non-negative (this is an extreme example to try and illustrate the point).

Rationale: Included in the proposed change.

Accidental under-generalisation that has arisen from my suggestion L367 (Comment 14) L367 – my previous suggestion to limit this to spring phenology was a mistake given the new information on the type of phenological events included being only 75% spring. But overall would consider removing this whole sentence for reasons given above.

Comment 17. Thank you for this change but the reference to the equator is still in the abstract, could that be changed as well?

Comment 27, not sure I fully understand the response. I've tried here to explain my point and interpretation of the response a bit more with a specific improvement suggestion.

Proposed changes: to acknowledge that any identified drivers will be proxies so do not need full biological realism (impossible really with an absolute window).

Rationale: To my knowledge of how sliding windows work, they start at a reference day and search back in time i.e. earlier than the reference. An example would be a reference day of 20th May then a 1 year search would be from 20th May Year1 – 20th May Year0. If the event was for example, breeding timing and recorded breeding was 24th May to 5th June across years, this would all be ok as the identified window was occurring before the event in all years. I find a bit of a problem with a temperature window for phenology occurring after the event has happened, which could occur if you set the reference day to the latest observation e.g. 5th June in the previous example. It could select a window of first week of June when most events have happened in May and therefore biologically could not have had any influence on those events. A causal relationship is impossible. However, this debate does not really matter that much, as all absolute windows will be proxies anyway and unlikely to be causal. However it is worth clarifying as sliding windows are often assumed as representing causal cues in phenology and could be misinterpreted in that field. For morphology, taking the latest date could make more sense, though, having the driver occurring before morphology is measured does still make sense. My key point is a driver window later than an event or measurement cannot influence it, one earlier can. But do let me know if I have misunderstood this.

New comment: Structure of Methods and Supplementary Material. Having re-read the Methods and Supporting information and possibly because of all the additions in the revision, I found some of the styles and structure hard to follow. I will try and be as specific as I can.

Proposed changes: more thought about the structure of the supporting information would be really helpful, I would recommend structuring to mirror the mentions in the main text or a standard paper i.e. intro related, methods related, results related, etc. I found the current structure with so much material and so many different models to be quite confusing to follow.

Explain more clearly how many meta-analytical models are fitted (maybe a table of them all) and justify the model structures e.g. why not include all important drivers throughout? Also, explain which model has led to the 'global effect' quoted and

how any other model structures impact this quantity (global effect).

Re-name the sensitivity section, include how these models were run (as this is the methods not results) and move results to supporting information.

Be clearer about the population size impact, or better yet, just include this variable in all models. If not the latter, justify why it is not included. Also correct Figure S18 as figure does not match interpretation text below i.e. graph says adding population size makes the effects significant, text says opposite.

Main areas of confusion:

1. How many models? It is not clear to me how many meta-analytical models were actually fitted and which are interpreted in the main text. In the methods in main text, it seems like an initial one with effects of study, location, and species as well as phylogenetic relatedness and also accounting for probability of being spurious and climate data quality, then one without phylogeny, then a third with the variables hypothesised to impact phenological responses (but not done for morphology?), then a fourth to test the phenological trait variable. I have a question as to why there are so many different models. Could the full model with variables hypothesized to be important not just have been used from the start and phylogeny influence checked then? It is not clear currently why so many model variants are being used in different parts and what impact this has on the results. Apologies if I have missed this, if I have, please just elevate mentions. Currently a reader can be left questioning reasons for choices.

2. Sensitivity analysis/Missing variables. This doesn't feel like the correct title for this section. It seems more like it is justifying not including certain variables in all models rather. I also typically consider sensitivity to assess impact of elements of the final model, not additional things that were not included. It was also a bit confused including both a missed variable and justification of not removing possibly spurious relationships. Furthermore, it reads as results not methods. It does not actually say how these models were run, just saying what their results were. It seemed a bit confusing here and should be in results? Or referenced in results but in supporting information and just referenced at relevant places such as L676-677 when the variable accounting for the probability of being spurious is mentioned.

3. Population size. It is clear that population size is highly important for G, so why was this variable not just included in the models interpreted in the text rather than being pushed to the supplementary? It seems that inclusion or not impacted significance of some effects, which is important but shrugged off here. I do not think it is currently helped that wording in the Supporting Information does not match the graph i.e. graph says adding population size makes the effects significant, text says opposite (Figure S18). But I wonder if there is an effect you know even a priori will be important, why not include it throughout? If there is a reason, it should be included. So, either switch the model or add a concrete reason why population size is omitted.

4. Structure of the supporting information. I would like to suggest structuring as 'Supplementary Methods', then Results, then Figures, or interspersing the figures where relevant. The current structure makes some things hard to understand. Some examples:

Figure S7 – should this not be first in figures as it refers to an initial step in the analysis? Also, Figure S10 possibly could go earlier.

Table S7 appears between several with results which feels a bit odd, could be first table as it is contextual or about data not results.

Moving methods before figures and tables of results would make them much easier to understand, e.g. Table S8.

Small details: Figure S5 mentions different sample size as SEMs for two taxon did not converge, was this also the case for the main analyses in main text. I might have missed it but this did not seem to be mentioned in relation to those analyses but should be. Figure S8 – don't jitter if you want to show comparability between two metrics. Also, why only for 35 studies? And how were they chosen? Figure S17 – what are the numbers? Are they sample size? It seems sample sizes might be more variable than has been mentioned, could this please be addressed. Section 'testing for non-linearity in relations' is very repetitive.

(Remarks on code availability)

As before, all good.

Reviewer #3

(Remarks to the Author)

(Remarks on code availability)

Reviewer #4

(Remarks to the Author)

Thank you very much for the extensive revision of the paper. The additional citations, elaboration in the introduction and details in the methods greatly improved the manuscript and addressed my comments.

Line 508-516: These sentences could be reorganized to improve the flow of ideas and reduce the repetitive phrasing.

Line 559-560: Can you be clearer as to why you think this is unfortunate? I assume you mean it is unfortunate that we are still biased in where most ecological time-series data are collected, but perhaps you mean something else about these studies specifically.

(Remarks on code availability)

Version 2:

Reviewer comments:

Reviewer #2

(Remarks to the Author)

I would like to thank the authors for their careful consideration of my comments and additions/changes to the manuscript. In particular, the wording of 'across-study effect', the tweaking of the conclusion statements, the clarification of models (Supplementary Table S8), and addition of motivation for morphology in the intro.

I also appreciate the clarification on Figure S11, I can now see that the figure and text do match. Though it seems the Figure has changed from the previous version so now only one effect is made significant by the removal of population size. I think it was the lack of circle for the PG effect that confused me before, I assume this is because it cannot be estimated without population size (which makes sense), but perhaps this could be stated in the legend just to make sure it cannot be missed by readers.

I also appreciate the time spent to explain the choice of reference day for the sliding window and the new figure.

I do think that the figure the authors presented in their response should be the one in the text. Showing the earliest date as well as the latest is very important, which I explain below. While I would like to emphasize that I think that the author's response is sufficient now for publication (thank you for this), I would still like to justify the use of the figure in the response. To do this, I have edited the authors' nice figure (attached as I cannot upload into this text box). What I have done is cut out the window around Oct (the shortest window in the figure) and move it to closer to the reference day. So, this should be a possible window that could be selected as the 'best' window. I hope this now shows the biological issue, which is that this window cannot have influenced egg laying for the earliest recorded events, as the window occurs after these events. It can correlate with them, but it cannot be causal for all years. Such an outcome may not have happened in any cases in this study (I don't know) but it could be using the latest event as the reference day. The response figure already starts to illustrate this as the 'best' window continues beyond the earliest event, but the Supplementary Figure does not. Therefore, including the schematic with the earliest recorded line will better illustrate this possibility to readers and will help correct interpretation of results, adding a possible window fully after the first date would be even better and more transparent. I would suggest a small wording change here, as a window that occurs after earlier events cannot be biologically 'relevant' in those years — though it may still correlate statistically. Thank you for the involved discussion on this, I do hope you will consider this point in any final adjustments.

(Remarks on code availability)

Didn't re-check this time but assume still good as before.

Reviewer #3

(Remarks to the Author)

(Remarks on code availability)

Response letter

Our references to the line numbers in the responses below are for the manuscript version (and SI) with the "Track changes" mode activated.

REVIEWER COMMENTS

Reviewer #1 (Remarks to the Author):

The authors do a great job addressing a timely question, by pulling together an impressively broad dataset and using a sophisticated multi-step analysis to arrive at a meta-analysis that teases apart direct and indirect effects of climate on vertebrate population growth rates. I am impressed that the authors were able to find so many species with concurrent population size and trait time series; I was only aware of a handful of these previously. The work should be of interest to a broad audience.

I think the simple SEM (conceptually simple, not so simple to implement) is the correct way to approach these data, and I am impressed by the care the authors took to propagate estimate uncertainties into the meta-analysis and the range of covariates that were used in the meta-analysis. I have no concerns about the analyses, and the script files are well annotated. I'm not at all surprised that the phylogeny didn't improve model fit, but it was worth testing anyway, and I applaud the extra work that went into its inclusion. I also appreciate the inclusion of the null models and intercept-only models as reference points.

Our answer: We thank the reviewer for the positive feedback!

Comment 1. Between lines 439 and 440, or somewhere else appropriate, please add the summary information from Fig. S16, that the median window duration was 2 or 3 weeks, for temp and precip respectively. I think that result belongs in the main text.

Our answer: As suggested by the reviewer, we have now added "The median window duration across the studies was 2 weeks when using temperature as predictor and 3 weeks when using precipitation as predictor (Fig. S9)." in the Methods of the main text (LL 577-578).

Comment 2. At L407, please add the range of grid sizes to the main text (0.05 to 0.5 degrees), so that readers don't have to check Table S6.

Our answer: Agreed, we have now included the sentence "The grid sizes ranged from 0.05 deg in Australia, to 0.1 deg in Europe and 0.25 in North America and 0.5 deg elsewhere." on LL 532-533.

Comment 3. The manuscript is written well. Here are a couple of minor suggestions for awkward places I spotted, but I had no major concerns about the writing.

L519: I'd use 'among studies' instead of 'between studies' (as you did on L600).

Our answer: Done.

Comment 4. L551: should be 'probability ... was' or 'probabilities ... were'

Our answer: Agreed, we changed to 'probability ... was'.

Comment 5. The order of panels in Fig. 3 is odd. Wouldn't it be just as clear to have panel a) in the upper left, and swap it with panel b)? I don't think the arrows showing relationship to axes are necessary. In panel a) it's suspicious that none of the phenologies are insensitive to climate, i.e. the blank gap around a slope of 0; the same gap is present in panel c of course. Is that gap a sign of publication bias? Other meta-analyses of phenological change don't show that gap.

Our answer: Thank you. As several reviewers pointed out that it is rather confusing than helpful to have the arrows between the panels pointing out the relationships, we have now revised this plot to organize panels in the conventional order, i.e. from a) to c).

The gap that you describe arises due to the use of sliding window analyses to identify the best climatic window for each trait. Since this analysis identifies the strongest signals, it dismisses weak relationships between climate and trait. We have now included a mention of this in LL 291-295 in the manuscript.

Comment 6. I'm also curious whether there is any phylogenetic signal to the variation in the slopes of panel a. Could those individual lines be color coded instead of all grey, to represent the groups of vertebrates (birds, mammals, fish)? A recent paper suggested there may be some phylogenetic signal there (Loughnan, D., S. Joly, G. Legault, H. M. Kharouba, M. Betancourt and E. M. Wolkovich. 2024. Phenology varies with phylogeny but not by trophic level with climate change. *Nature Ecology & Evolution* 8(10): 1889-1896.).

Our answer: Thank you. We did colour-code this figure but as majority of the studies are on birds find it not very informative. Therefore, we opted for including this figure in the Supplementary Materials (new Supplementary Fig. S4). We have also re-run the meta-analyses per taxon now, whenever the sample size allowed doing so: for birds ($n = 82$) and for mammals ($n = 7$). We briefly mention (LL 336-340) now that "The across-study patterns in phenological responses to temperature_d (CZ), effects of phenology on G while accounting for climate and population size (ZG), phenology-mediated effect of temperature_d on G (CZG) and direct effect of temperature_d on G (CG) seemed to be predominantly driven by birds, which constituted most of our dataset." and include the results of these taxon-specific analyses in Supplementary Fig. S5. We also now include a paragraph in the discussion (LL 427-444) in which we compare our results on the degree of phylogenetic structuring estimated when using the full dataset and the subsets focusing on birds and mammals and discuss our findings in the context of the previous literature (e.g. Loughnan et al. 2024).

Comment 7. In Fig. S8 some of the models have extremely high R² values. Is this potentially because in Fig. S9 many of the species have very strong density dependence (PG effects are much larger than any others)? While density dependence is biologically important, it would be nice to know how much of the high R²s is due to the PG arrow rather than CZG or CG.

Our answer: Indeed, density dependence is a key process driving population dynamics. As suggested by the reviewer, we have now partitioned the variance in G explained by C (temperature), Z (phenology) and P (population size) in “the population growth model” that is included in our SEMs (see Methods). As expected by the reviewer, the largest proportion of variation in G is typically due to population size (median = 0.26, 5th percentile = 0.01, 95th percentile = 0.65 across studies). Phenology explains a lower proportion of the variation in G, although in some models it can be quite high (median across the models = 0.05, 5th percentile = 0.001, 95th percentile = 0.43 across studies). Finally, the direct effect of temperature explained a median 0.03 of the variation in G across the models (5th percentile = 0.001, 95th percentile = 0.42). We included these results briefly in LL 697-699 and, in more detail, in the Supplementary Results and as Fig S21. In the Supplementary Results we also clarify that “The contribution of temperature to population growth rate is only partial here, as it only captures the direct effect of temperature_d on G (i.e. we applied variance partitioning to G). The effect of Phenology on G includes an indirect effect of temperature_d (CZG pathway), and we note that a large proportion of variation in Z was explained by C (see R² of the “trait model” within our SEM, Supplementary Fig. S12).”

Comment 8. In Fig. S9 it would be nice to see which effect sizes for the different arrows belong to which species. I’m wondering if lines that connected the points within a column would reveal patterns of covariances among those, for example species with higher CZ would also have higher ZG, or vice versa? Of course, its also possible that that many lines would be a horrible visual mess and not lend any insights, but it may be worth checking. If there are any strong covariances among those (which would be easy to check), that would be worth mentioning in the main text too.

Our answer: We had tried several different ways to show the relations between CZ and ZG on former Fig. S9 but that unfortunately led to very complex figures that were difficult to read and looked rather ugly. We also note that we already show the relationship between CZ and ZG for each study in Fig. 3d and panels c in Supplementary Figures S26-S28 (focusing on other combinations of phenotypic traits and climate variables). We think these existing figures are the best and most direct way of visualizing this information. As suggested by the reviewer, we have now calculated Pearson correlation (i.e. standardized covariance) between CZ and ZG. For the dataset looking at phenological responses to temperature this correlation is positive and significant, and we mentioned this in the main text (LL 300-306), as suggested by the reviewer. We report the correlations for other combinations of phenotypic traits and climate variables in the legends to respective figures in Supplementary Information (Figs. S26-S28).

Comment 9. In S16, can the length of the window be related to generation time, or something else about the pace of life? I’m surprised by the length of the longer windows.

Our answer: We have checked whether there is a relation between window duration and generation time, but found none: for temperature windows the correlation between their duration and generation time was Pearson r ($df = 155$) = -0.124, $p =$

0.12; and for precipitation windows the correlation between their duration and generation time was Pearson r ($df = 159$) = -0.02, $p = 0.82$. See also the Fig. 1 below.

Figure 1. Relation between generation time (in years) and window durations (in weeks) for temperature and precipitation variables.

Reviewer #1 (Remarks on code availability):

The data are accessible, and the scripts look to be well annotated. There are MANY script files, and I did not open all of them. The ones I checked looked good.

Our answer: Thank you for your positive feedback.

Reviewer #2 (Remarks to the Author):

This paper uses path analyses and meta-analytical approaches to explore phenotypic trait mediated climate impacts on population growth in vertebrates. They used 213 time series of phenological and morphological traits to test the path of influence of temperature and precipitation effects on population growth. They found highly variable results across the species and locations. In general, phenological events were more climate influenced than morphological traits, which showed no clear trait-mediated climate effects. The most noteworthy results are that phenology was predominantly temperature driven with then positive impacts on population growth, suggesting adaptative phenological responses to temperature change in many species. This effect of phenology was however shown to be weaker at lower latitudes. Additionally,

morphology was insensitive to climate changes.

Here, I begin with some overarching summaries covering topics requested in the reviewer guidelines, before giving a detailed review.

This study provides some original insights into the topic of climate change impacts on animal populations growth, thus giving an important contribution into the field. It is a novel study researching trait-mediated effects of temperature at a global scale and for multiple populations. The findings from this work will be of significant value in the field and related fields in demonstrating why researchers need to consider different pathways of influence of climate effects. It is also a pertinent reminder to consider population level impacts. The methodology is sound, and I would say exceeds expected standards in the field.

Our answer: Thank you for your compliments and positive feedback!

The paper is well written, easy to follow, and has a good flow. The methodology in particular explains some very complex and detailed analyses in a clear and concise way, and I would like to complement the authors on this. The data analyses used in this paper are really clearly written and explained. These are complex analyses but well thought through and supported with a well-documented R package, which is to be commended! I would especially like to highlight the treatment and propagation of uncertainties as a really nice part of the method.

Our answer: Thank you for the positive feedback. We are happy to hear the methods are easy to follow and hope they will be easy to replicate in the future.

Comment 1. However, the discussion of the previous literature to support the context and claims (primarily in the introduction and discussion) could do with improvement. Specifically, the taxonomic scope of the references is quite limited, particularly to bird species, and given the broad scope of the paper, this should be reflected in the cited literature. Examples of particular areas where this could happen are detailed below but I would recommend reviewing the whole introduction and discussion with this in mind as well.

Our answer: Thank you for highlighting this skew in the cited literature. We have now revised the Introduction and Discussion carefully and paid attention to integrate more diverse references, especially regarding their taxonomic scope. As a result of expanding the cited literature, our reference list has now 27 new added resources. We answer your detailed comments below.

Comment 2. Following from this, the work conducted does support the broad conclusions, however, there is some over-generalisation of some of the conclusions, e.g. not take into full account the limitations of the (granted very good) dataset e.g. geographically and taxonomically biased. Please also see other comments about using averages to draw conclusions from highly variable inputs and results.

There are a few areas where I believe this manuscript would benefit from addressing, which I detail below. Any consideration of the points mentioned would require minor revision and should not prohibit publication.

Our answer: Thank you very much for your constructive comments. We have revised the manuscript to address the issues raised by the reviewer, e.g. to extend the referenced literature beyond birds and to underline the limitations of our dataset. For more details see below the response to each of your specific points.

Main areas for improvement:

1. **(Comment 3).** Expanding literature support, particularly increasing taxonomic scope of cited literature. There are also some cases where claims made are not currently sufficiently supported.

The authors could, for example, pitch their study importance more effectively in the introduction, by highlighting how the importance of looking at population growth effects has been previously underscored in the literature, yet there is a need for more studies to test this (e.g., Samplonius et al., 2021). Samplonius, J.M., Atkinson, A., Hassall, C., Keogan, K., Thackeray, S.J., Assmann, J.J., Burgess, M.D., Johansson, J., Macphie, K.H., Pearce-Higgins, J.W. and Simmonds, E.G., 2021. Strengthening the evidence base for temperature-mediated phenological asynchrony and its impacts. *Nature Ecology & Evolution*, 5(2), pp.155-164.

Our answer: Agreed. We have extended the cited literature, trying to cover as broad range of species as possible. We also follow the suggestion of the reviewer and highlight the previous research (Samplonius et al. 2021) showing that population sizes must be recorded in parallel with phenotypic traits to allow for a better, mechanistic, understanding of climate-driven changes (LL 154-156).

Comment 4. L295 – This is a good example of lack of support, reference 34 is one example of one species and one phenological trait that is being used to support a statement that phenological traits in general are plastic. Either change the claim or add greater support from a wider range of taxa and phenological events. (There are many other examples like this and I will not have found them all).

Our answer: Agreed. We have now added additional references that range broadly in the species they studied and regarding the studied phenological event (LL 377).

Comment 5. L312 – again a single reference of support to a very general claim.

Our answer: We have added additional references to support this claim (LL 396).

Comment 6. The support for the use of morphological traits and particularly those chosen, several of which seem that they would not change within life beyond developmental changes e.g. snout-vent length, is not sufficient.

Our answer: We have now extended the list of references, also citing studies that showed responses of other, more labile, morphological traits, such as body mass. Further, we tried to cover as diverse taxa as possible: e.g. phytoplankton, zooplankton, plants, birds and mammals.

Comment 7. e.g. L184 – not many references chosen and not linked to morphological traits included.

Our answer: We extended the list of references, see our responses to similar comments (comments 1, 3-6, 8) above and below, see LL 208-210 in the revised manuscript.

Comment 8. L186 – could not see any supporting references for morphological traits beyond body mass and body condition (which is actually calibrated using body length, further suggesting this trait should not be considered to be labile or plastic). Support for snout-vent length, body length, and bill length (another which I'm not sure should change interannually in response to weather conditions) was not given.

Our answer: We have now extended the list of the studies we refer to, including the papers that looked at body mass as well as at more structural morphological characteristics (such as snout-vent length, body size, bill and wing length) and trying to cover as diverse taxa as possible (LL 208, 210). We appreciate the reviewer's point that often body size metrics are considered to be less plastic than for example body mass within a lifetime of a species with determinate growth. However, recent studies, now cited in our revision, have shown that mean population values in body size metrics can vary considerably from year to year. Such short-term interannual variability in body size can be due to labile plasticity within a lifetime (e.g wingspan depends on annual wear and investment in feather growth after moult). It can also be due to developmental plasticity causing variation among generations, which in turn will affect mean population values in body size metrics from year to year due to the new cohorts entering annually. We thus think there are sufficient reasons to investigate how body size metrics may vary from year to year and how this is affected by climate.

2. **(Comment 9).** Consider whether the averaging of results in the meta-analysis is a meaningful metric to assess the results.

Averaging the path coefficients to give (what I assume is a mean) result, I feel, is not the most meaningful metric. This is particularly the case given the different phenological or morphological traits being considered and due to the variability of results where a mean of 0 could result from all results being close to 0 or from as many above 0 as below. By saying that the average of 0 indicated adaptive responses does not give more insight than looking at the number of non-negative results but it does perhaps have some biological issues. The traits being considered are quite different, it is perhaps not to be expected that things like lay date would have the same response and effect as emergence date or parturition, in the same way body mass would not be expected to respond in the same way as body length or bill size. Therefore, comparisons among these responses or analysis of modal or majority trends are ok but direct averaging ignores these differences.

Our answer: Meta-analysis leverages the effects reported in multiple studies (often studies that lack power) to assess a general, "global effect size", across the studies. This was exactly our aim in this study: to reveal the general response of traits to climate in vertebrates and to assess how these climate effects on traits propagate to population growth rate. Even though it is tempting to think of a meta-analysis as averaging across

the studies, we note that the studies with larger sample sizes will contribute more strongly to the global effect size than those with smaller sample sizes, all other things being equal.

With regards to reporting the majority or modality trends: such counting of positive or negative effects is against the whole philosophy of meta-analysis of borrowing the information from each study to be able to say how general the response is (see e.g. (Koricheva et al., 2013)). Therefore, we decided against reporting such counts.

We do agree with the reviewer that the variability across studies is at least as interesting as the mean across studies. Therefore, in the original version of the manuscript, we already analysed how the variation among studies in effect sizes (a) depended on species characteristics (in the section on 'Explaining among-study heterogeneity in climate effects'), and (b) varied between different types of traits.

This comment (and one of your comments below) made us realise that some of these analyses and their findings were perhaps too much hidden in the Methods section of the manuscript (formerly on LL 596-603). Therefore, we have now included these findings in the main text (LL 340-347), to highlight the comparison in responses of different trait types, as suggested by the reviewer.

Comment 10. L233-236 – this is an example where I do not find the averaging so helpful as it you may not expect all traits to behave the same, and a lack of an average response in a particular direction does not mean that for individual species and traits that they are not adaptive.

Our answer: Thank you. This sentence states the general expectation of our study. We think formulating such expectations is very useful and helpful and the hypothesis-driven research was based on them. Moreover, this specific expectation was tested with the binomial test (see LL 679-684) which is not based on averaging.

Comment 11. L295-297 – again, not sure about the averaging particularly given the highly variable results. I would find it more meaningful to look at majorities of directions or between group differences etc. The average does not really say anything biologically.

Our answer: Please see the response to your comment 9 above. We hope it clarifies the difference between meta-analysis and averaging and why simple "vote counting" is against the underpinnings of meta-analysis.

Regarding the specific lines to which the reviewer refers, this text emphasizes that our findings suggest that phenotypic plasticity seems to be the mechanism underlying the observed effect of temperature on population growth rate as mediated by phenology. Indeed, these lines refer to our prediction that CZG close to zero is expected to be observed if "trait changes owing to adaptive phenotypic plasticity would allow accurate tracking of changing climate while avoiding possible detrimental effects of climate on population growth, such that ZG is close to zero (and hence so is CZG)". (LL 269-271)

3. **(Comment 12)** Reduce over generalisations. E.g. saying phenology when mean spring phenology.

This study only looks at spring phenological events (as far as I can tell) but this is not mentioned anywhere. This needs to be mentioned from the very beginning as it strongly impacts the interpretation of the results. It is not a drawback, just a clarification that is important.

Our answer: Thank you! The reviewer is correct, most of the studies (75%) focus on spring phenological events. We now clearly stated at the beginning of the Results section (LL 231-233) that most of our data are on spring phenology. We also extended our Supplementary data with a column that specifies the season in which events are observed for each study.

Comment 13. L141 – this is certainly one example of the impact of phenology; however, it is very specific and it does not hold for all biological events e.g. those in Autumn or if caterpillars hatch prior to their food source. Please be more specific about the scope of events you cover – just spring phenology in a few species. The caterpillar example is still spring but would not always fit this pattern.

Our answer: We are not sure what the reviewer means by “caterpillars” in the context of the text on former L141 (and this whole paragraph). This paragraph transmits the message that looking at how changes in phenotypic traits affect fitness components tells us just part of the story and we argue it may also be useful to look at how trait changes relate to population growth rate. In this whole paragraph we do not mention caterpillars and the paper we refer to in the first sentence (Radchuk et al. 2019 Nat Comms) looked at adaptive responses across animals. Sorry, if we misunderstood this comment, we are happy to consider it if we receive more specific explanation.

Regardless, we have now carefully revised the text throughout the manuscript to clearly state that the majority of the studied phenological events are spring ones, as suggested by the reviewer and detailed in our answer to their comment above.

Comment 14. L289 – this is a bit over generalised due to the limitations mentioned below, in particular geographic and taxonomic biases. In addition, it is only a subset of spring phenology that has been looked at. I suggest moderating the language here to focus on spring phenology, clarify that vertebrates are mostly birds, and across most of the globe (not all). Given that you have results for individual studies – can you say for which groups these temperature mediated effects are most clear?

Our answer: Done. We have now revised this sentence to account for the reviewer’s suggestion. It now reads: “Here, by using a large set of species and studies, we demonstrate that spring phenology mediates effects of temperature on population growth in vertebrates found across most of the globe. Our results are predominantly driven by birds, because the relevant data were mainly found for this taxon despite our initial intention to address this question for vertebrates broadly” (LL 367-371). Further, we have added the results of the analyses run separately for birds and mammals to Supplementary Figs. S4 and S5. We could not run a meta-analysis for reptiles only because of a too small sample size ($n = 4$). We also mention now in the Results that

the across-study results were mainly driven by birds, which constitute most of our dataset (LL 336-340).

Comment 15. L596-603 – I find this a more specific result and interesting, I think this should be elevated to the main text not hidden in the methods.

Our answer: Agreed, we have now shifted these findings to the main text (LL 340-347), see the answer to your other comment above.

4. **(Comment 16)** Include more comment or a specific section on the study limitations. e.g. geographic spread, taxon spread, type of morphological traits chosen – possible not plastic or labile (some of the references they have use body length to standardise body mass to look at body condition). Also include the different types of phenology and morphology and whether they can truly be compared.

Our answer: Agreed, we have now included a new paragraph in the discussion (LL 457-468) that is devoted to the limitations of our dataset, as suggested by the reviewer.

Comment 17. The geographic limitations of the dataset need to be acknowledged early, especially the lack of studies in the tropics and from low latitudes and the bias to Europe and North America. This is particularly important given the latitude effect that was found, some comment on whether this effect would hold into the tropics or with more data would be good. E.g. L283 it mentions close to the equator, but it seems there is limited data in this study from the equator.

Our answer: Agreed, we now point out the geographic bias in the assembled dataset at the very beginning of the Results section (LL 235-237) focusing on the dataset description. As suggested by the reviewer, we now also highlighted this geographic bias in the Discussion (LL 460-464). In this passage we warn that our findings remain to be tested in the future, when the studies covering latitudinal gradient more systematically are available.

For the specific point regarding the former L283, we have now revised the text to “The ‘direct’ effect of temperature_d on G that remained after accounting for the phenology-mediated effect of temperature_d on G was positively associated with absolute latitude, such that it changed from being negative at lower latitudes to around zero towards higher latitudes” (current LL 360-364).

Comment 18. Another potential limitation of the study is the substantial variation in climate data resolution across the countries considered. Specifically, the resolutions for North America, South America, and Africa are relatively coarse. Such large-scale temperature data may not accurately capture the environmental conditions experienced by animal populations, particularly the microclimates to which they respond. Please include a small mention of this.

Our answer: We mention this in a new paragraph (LL 457-468) that we added to the discussion to focus on the limitations of our dataset (see also a response to your comment above).

Comment 19. L318 – this comes in very late and is not fully explained. What does this mean for all other results?

Our answer: We have added more details now to better explain what is meant and the possible implications for other results: “Our dataset comprised 97 time series, which is not a small sample per se, but because we relied on previously published data, we had no control over the distribution of the species characteristics within this dataset. Therefore, future extensions of the dataset aimed at covering a wider range of species characteristics would allow for better addressing of this particular question.” (LL 405-409).

Comment 20. L399 – 402 – I feel that these limitations need to be mentioned further up somewhere.

Our answer: Thank you. As already mentioned above, we shifted all of this information to the very beginning of the Results section that describes the dataset (LL 226-237).

Minor comments:

General

Comment 21. L130 – I would replace ‘global warming’ with ‘climate change’, I appreciate it is said a few times in these sentences but it is a more accurate characterisation of the actual changes that will be experienced.

Our answer: Changed accordingly.

Comment 22. L311 – the population is not plastic or not, the trait is.

Our answer: We have now revised this sentence to read “Another is that the traits of lower-latitude populations are in general less plastic and hence these populations have narrower thermal tolerance than higher-latitude populations⁶¹⁻⁶⁴, who typically experience higher intra-annual temperature variation.” (LL 394-397)

Comment 23. L312-315 – I do not agree with the reasoning here. Surely physiology mediates or plays a role in all traits, particularly phenological traits and therefore, it will be playing a role at all latitudes. I do not see physiology and phenology as separate, physiological processes will be the basis of the phenological events we observe (e.g., gonadal development for laying date). Perhaps you could be more specific about which physiological processes you think are influencing things at lower latitudes. Or perhaps the authors are referring to physiological effects as direct temperature effects and phenological effects as indirect temperature effects (e.g., mediated through food supply). If so, then perhaps this distinction could be clearer.

Our answer: We have now deleted this sentence as this paragraph focuses on latitudinal effect and the mention of physiology was out of place.

Comment 24. L344 – again, not sure about this distinction, could you be specific about the physiological traits. Also again, phenological traits will be built up from physiological

traits and indeed in some species they could be the same e.g. trees and invertebrates. Perhaps add a qualifier about vertebrates here.

Our answer: Thank you. We have now revised as suggested, the sentence reads: "As changes in both morphology and phenology arise from changes in underlying physiological traits (e.g., gonadal development for laying date)⁶⁷, we expect that physiological traits would respond much faster than phenological or morphological ones and likely mediate many climate effects on populations." (LL 451-455)

Comment 25. L352 – what about empirical models? I do not think this is a clear distinction as models can be applied to empirical datasets, did you mean theoretical or simulation studies?

Our answer: Thank you for spotting. We have now revised to "whether based on observational, experimental or modelling studies" (LL 474-475).

Comment 26. L380 – this link does not work – takes me to a 404 page

Our answer: The link is updated.

Data and methodology

Comment 27. More detail on how the biological realism of the identified climate windows would be a nice addition. While searches like sliding window can identify climate drivers that make excellent explanatory and predictive models, they are not always biological realistic. In this paper some choices around the window searching could lead to biological realism being reduced, for example, 2-year search could actually exceed the lifespan of some short-lived species in the dataset, or at least be before they are born when they reach maturity at age 1. In addition, the use of a reference day that is the latest date a phenological event was observed will in many years be occurring after the event takes place. Some discussion of the biological realism and implications for interpretation would be a good addition.

Our answer: In our dataset only 5 species (7%) have generation time ≤ 2 years (see the histogram of generation times in the Fig. 2 below). A fair share of the species in our dataset are long-lived species and for them being able to account for potential carry-over effects of the previous years is important compared to the very low proportion of species whose generation time is shorter than 2 yrs.

We have now included text in the Methods section explaining in more details our choice of the 2 years period prior to the biological event for the sliding window analysis: "Our dataset includes diverse species, ranging from birds to mammals, reptiles and fish. For many of these species biological knowledge on what constitutes a relevant climatic window is unavailable. Therefore, the use of sliding window analyses allowed to assess the climatic window in a standardized way, systematically, across all studies and species. Our identified climatic windows for passerines correspond to the windows that are known to be important in driving egg-laying of these birds based on their long-term studies, suggesting that our approach is valid. Our consideration of sliding windows up to two years before an observation may include potential climatic windows that are not very meaningful for short-lived species (those whose life span is < 2 years). However, only five of the species (7%) in our dataset have generation time

<= 2 years, justifying the use of the two-year period to capture potential carry-over effects for most of the much longer-lived species in our dataset.” (LL 563-574).

Regarding the use of the latest date that phenological event was observed over the whole study period: it is needed to not miss the event if it, as the reviewer correctly points out, occurs earlier in the year. Had we used the earliest day as a reference day, we would not be able to correctly identify the windows in those years when the event took place after that day.

Figure 2. Histogram of the generation times of species in our dataset. The red vertical line shows the median across the species (7.9 years).

Comment 28. Why was the detrending of the climate variables done after the sliding window analysis not before? I am curious as the given rationale would seem to hold for detrending prior to the sliding window.

Our answer: That is because prior to performing sliding window analyses it would be unclear what data to detrend as the climatic window is not identified yet. In other words, we first need to have a climatic variable value for each year in order to be able to then detrend it over time. This comment made us realise that the fact the climatic variables were year-detrended in our analyses was perhaps not very clearly stated in the original submission. Therefore, we have now revised the text to state this early on, already in the end of the Introduction: “In all analyses we considered year-detrended temperature and precipitation (i.e. residuals) to avoid potential spurious effects caused by other environmental temporal trends^{46,47}.” (LL 210-212) Further, we now used ‘temperature_d’ in the methods and results sections as well as on the axes labels (Fig. 3) to refer to the year-detrended temperature.

Comment 29. L257 – p-values are not typically indicative of trend directions alone (could be in a one tailed test of some kind). Is this what is meant here, or did you mean beta value? If you did mean p-value, maybe explain how it indicates the trend direction.

Our answer: Agreed. We have now revised to read “..., with a tendency towards negative direct temperature effects on population growth.” Indeed, you are right, we referred to the negative beta value that was in the original manuscript reported on LL 254 and is now found on LL 314-316.

Comment 30. L406 – please mention that the resolutions vary here.

Our answer: We have now added “The grid sizes ranged from 0.05 deg in Australia, to 0.1 deg in Europe and 0.25 in North America and 0.5 deg elsewhere.” (LL 526-527), also to address a similar concern raised by the Rev 1.

Comment 31. L438 – please provide these preliminary analyses somewhere like SI.

Our answer: We have done so now, by including the new Supplementary Fig. S8 that shows the relation between window durations computed using daily and weekly resolutions, for a set of 35 studies.

Comment 32. L537 – is the difference to 0 something you would expect?

Our answer: This is an excellent point. Yes, as outlined in our expectations (LL 261-281), CZ would differ from 0 if phenology has advanced with temperature (negative CZ values) or if it was delayed with increasing temperatures (positive CZ values). Testing whether ZG differs from 0 allows discerning whether phenotypic responses to temperature are likely due to plasticity. The ZG that does not differ from 0 implies a plastic response and ability of individuals to respond to changing temperatures by adjusting their phenology (with no association between phenology and population growth rate for each given temperature). Further, we would not expect 0 necessarily for CZG, as explained on LL 267-274. For CZG we actually rely on a binomial test to assess whether the proportion of studies with non-negative CZG was higher than by chance only. To clarify this, we revised this sentence as follows: “To test our expectation that phenological responses to temperature allow avoiding temperature-induced population declines, we assessed whether the proportion of studies with non-negative CZG was higher than by chance only. For this we used a binomial test and tested whether the proportion of studies with non-negative CZG differed from 0.5 for a given combination of climate variable and trait category.” (LL 679-685).

Regarding the tests of whether CZ and ZG differ from 0: these allow to decompose the overall CZG path into phenological sensitivity and the association between phenology and population growth rate (given the temperature and population size, see our conceptual diagram Fig. 1). We think that such a decomposition is useful as it breaks down the overall path of phenology-mediated impact of temperature on population growth rate (CZG) into its underlying components.

Comment 33. L540 – could you give a bit more detail on the binomial test please e.g. I assume it has an assumed probability of some kind?

Our answer: Indeed, as the text says, “To test our expectation that phenological responses to temperature allow avoiding temperature-induced population declines, we assessed whether the proportion of studies with non-negative CZG was higher than by chance only. For this we used a binomial test and tested whether the proportion of studies with non-negative CZG differed from 0.5 for a given combination of climate variable and trait category.” (LL 679-685). We hope it is clear, that we compared the proportion of the studies with 0.5 and that 0.5 reflects the expectation when the pattern would be due to chance.

Comment 34. L579 - the same way as what? The reference here is not clear.

Our answer: We referred to the previous sentence. To clarify now we have extended the sentence to say “Specifically, we used the same random effects as in the mixed-effects models used to infer across-study effects (see “Across-study inferences”) and propagated uncertainty in the same way. We additionally included as predictors the variables hypothesised to affect the phenological responses to temperature...” (LL 729-731).

Conclusions

Comment 35. See above on generalisations.

Our answer: Done. See answers to your comments 12, 14, 16, 17 above.

Figures

Comment 36. Figure 1 – Conceptual diagrams need to be really clear. The overlapping of arrow heads with boxes makes this less so, could you add just a tiny bit of breathing room around each arrow head please. Esp in panel a under the top box.

Our answer: Changed as suggested.

Comment 37. Figure 3 – The order of the panels here is not very clear. Perhaps the letters could be changed to match the order – that makes it easier for the reader to assimilate the information. This figure also needs tidying, the arrows are different sizes and the panels are not full aligned.

Our answer: We have removed the arrows from panels a) and b) to the panel c) and now use the conventional order of the panels and letters. We have also adjusted the figures as suggested.

Supporting information

Comment 38. Supplementary figure S9: and any similar figures. I find the colour and shape differentiation very hard to see. Please choose some that are more distinct. At the size they are, even on a big monitor, I cannot easily see the difference between the circle and the square, nor light-dark blue/red.

Our answer: We modified the colour-scheme, and the symbols used in this figure as well as in former Supplementary Figs. S12 and S13 (current Supplementary Figs. S14, S17, S18).

Comment 39. L313 – I found this conceptual framework hard to follow and it did not seem to fit with the results in the main text. I hope I did not miss something, if I did, please correct me. But especially given that the climate data were detrended in the main text, I do not see how these two scenarios were tested. Scenario is also not mentioned in the main text. I suggest removing this section.

Our answer: Thank you, we have now removed this section from the Supplementary material. Indeed, there is not much mention of this in the main text. This is a simulation study that we have done prior to performing the meta-analysis, to formulate our expectations about CZG and CG. But we realise now that it may be more confusing than helpful and therefore have deleted this section.

Comment 40. L340 – GR is not defined.

Our answer: We completely deleted this section following reviewer's advice, this is obsolete.

Reviewer #2 (Remarks on code availability):

Comment 41. I have looked through the repository but have not tried to run all of the code. The repository is well organised with a nice ReadME to begin and all code looks great. The functions I have looked at are really comprehensively commented and clear.

Our answer: Thank you for your positive feedback. We do hope that the repository is structured and documented well enough to allow those interested to replicate the results or to apply this framework to the new datasets. The repository does contain many scripts as our framework required a multi-step approach with several different analyses.

Reviewer #3 (Remarks to the Author):

Reviewer #4 (Remarks to the Author):

In this manuscript, the authors Radchuk et al. conduct a meta-analysis using time series of vertebrate phenology, morphology, and population growth. Using this diverse data, the authors test the relationships between temporal and morphological traits, climate, and population dynamics. Morphological traits showed no relationship to temperature or precipitation and population trends, but a relationship was found between temperature and phenology---but not phenology and population responses. These results suggest that other traits are shaping population responses to climate change than those represented in the current literature.

Comment 1. Understanding the relationships between traits and population trends is important, as it may provide insights needed to forecast and manage species under climate change. Synthesizing trends globally and across diverse species assemblages also allow us to identify general trends and large-scale spatial patterns. The dataset presented in this study is impressive, particularly given the efforts to collect raw data from the original authors. However, I was surprised that the phylogenetic signal was weak. There is strong evidence of phylogenetic conservatism in phenology, with the variation due to latent traits potentially reflecting traits favourable under ancestral climates. While I appreciate you did include a phylogeny in your analysis, there was no justification or discussion of the importance of evolutionary history in the introduction. I would have like more detail on exactly how phylogeny was included in the model, whether specific packages were used, and what the estimated phylogenetic structure was. As an additional supporting analysis, I am curious whether relationships would be stronger within the individual taxonomic groups, especially given the diversity in the types of phenological events included in the study.

Our answer: Thank you for this comment. As suggested, we now point out on LL 241-244 that "Since there is evidence of phylogenetic structuring for phenological events⁴⁶ and evolutionary history may potentially shape trait-mediated effects of climate on populations, we accounted for phylogenetic relatedness in our meta-analytical models." We explain how phenology was included on LL 659-662: "...these models also accounted for the phylogenetic relatedness by allowing the values of the random species effect to be correlated according to the phylogenetic correlation matrix that was derived from the phylogenetic tree⁷⁷". We have thus used phylogenetic distances between the species in the dataset to account for potential correlation in the random structure of the respective effect sizes. To our knowledge that is how phylogeny is commonly accounted for in such kind of analyses, see e.g. Cinar et al. (2022). We fitted these analyses with the R package metafor, as mentioned on LL 686. We also now report the estimated Pagel's λ values on LL 328-335 and in Supplementary Fig. S3.

As requested by the reviewer, we have now re-run the meta-analyses separately per taxonomic group, specifically for birds ($n = 82$) and mammals ($n = 7$). The sample size for reptiles was too low ($n = 4$) to fit phylogenetically-corrected analyses. The findings for these additional analyses are reported on LL 336-340 as well as in the Supplementary Results and Supplementary Figs. 5 and S22-25. We have also now devoted a separate paragraph in the discussion to interpretation of these findings and discussing them in light of previous research (LL 427-444).

Comment 2. The framing of the project and writing could also be improved in my opinion. I felt the state of the literature could be better represented and some of the broad statements tempered. I disagree that no work has been conducted on this topic before. But perhaps more elaboration and nuance would address this, as I do believe that there are few robust and global studies on this topic and appreciate the novelty of these large-scale studies.

Our answer: Thank you. A very similar point was raised by another reviewer, and we have now revised the Introduction carefully to tone down the statements as well as to broaden the cited literature.

Comment 3. The discussion of the results would also benefit from more nuance, as the lack of statistical significance was at times not clear from how trends were discussed. Finally, the discussion would benefit from more discussion of future directions and applications of these findings to conservation or species management.

Our answer: We have now revised the discussion carefully, paying particular attention to stating clearly what findings were significant or not. We also added a new paragraph that discusses the limitations of the current dataset and specifies the directions for future research efforts in order to more systematically cover different taxonomic groups and geographic regions (LL 457-468). We also explicitly stated in the discussion that “we may need to embrace ecological complexity by studying multiple phenotypic traits and considering predictors that capture intra-specific variation.” (LL 487-488). We refrained from discussing the applications of our findings to conservation and management because we find it is not straightforward yet to give conservation recommendations to mitigate the impacts of climate change, this will become more feasible once the field expands and matures. Further, as our dataset covers species that differ widely in their biology and habitat requirements, we think that such general statements are rather difficult to make, and recommendations should be species-specific.

Specific comments:

Comment 4. The title should more accurately reflect the study results. The current title makes it seem like the relationships between temperature and precipitation were strong, but in reading the text, it was only temperature.

Our answer: Agreed. We have modified the title to “Changes in phenology mediate vertebrate population responses to temperature globally”.

Comment 5. Line 110: More context for your study is needed here, it sounds like it was exploratory, but I am sure a lot of thought when into why you collected all these time series.

Our answer: We are not quite sure what exactly the reviewer means. Our research is not exploratory, as is explained in the Introduction, we aimed to assess to what extent the effects of climate on populations are mediated by phenotypic traits, across species. The requirements of Nature Communications to the word count of the abstract are rather strict (150 words). Therefore, we tried to succinctly highlight the research gap that our study addresses already in the first sentence. That is why, the second sentence (the one to which the reviewer refers to) directly explains what kind of data were gathered to address that research gap. We do not see how we can give more details while sticking to the word limit, but we are open to suggestions.

Comment 6. Line 114-119: I think these sentences could be edited to better guide your reader and make it clearer that you are testing several relationships and why.

Our answer: Please see the answer to the above comment 5. The abstract is limited to 150 words, and we thus cannot afford to explain in detail all the relationships that we have tested and why we have done so. Instead, we opted to directly present the findings.

Comment 7. Line 127: This introductory paragraph introduces some big topics that are not fully developed. Could you break this into multiple paragraphs and further develop some of these topics?

Line 132-135: This should be its own paragraph to convince the reader this is important and true.

Our answer: Thank you. Surely this text can be “decompressed” and presented in two or three separate paragraphs. But we here intended to “come to the point of our manuscript” as quick as possible, so that the reader knows rather quickly what is the main topic that we are dealing with. We think it is rather a matter of taste and of course there are different approaches to writing. We did revise the Introduction and Discussion to explain more in detail some specific points and to pitch our study better, as suggested by this and other reviewers, but we decided against expanding the first paragraph into three paragraphs.

Comment 8. Line 141: Earlier phenologies can have huge costs as well, your argument would be stronger if you include some of these examples.

Our answer: Thank you. We have now revised this sentence accordingly: “Earlier timing of biological events in warmer years is associated with fitness benefits on average¹⁹ but it may also have huge costs by exposing individuals to extreme events, such as cold spells^{20,21} .” (LL 149-151).

Comment 9. Line 145: What is meant by “net demographic consequences of interannual variation in phenotypes”? Is there not an existing literature of traits being included in integral projection models to do exactly this?

Our answer: We have reworded it now to “Indeed, whilst many studies have examined the consequences of trait change on single demographic rates (e.g., reproduction, survival) at the individual^{16,23} or population level^{4,6,21,24}, few have examined the consequences of climate-driven changes in traits on the net demographic consequences of interannual variation in phenotypes that is driven by climate on interannual variation in population growth rate^{8,11,12,14,25}.” and hope it reads better. Also, among the examples of studies that did look at how climate-driven changes in traits affect population growth rate we have now included these IPM studies: Simmonds (2020 ELE) and Coulson et al. (2011).

Comment 10. Line 150-152: Could you elaborate and give an example of this? How common is this actually?

Our answer: We have now given more details, as following: “Studying the consequences for population growth rate is important because changes in fitness components (e.g. survival, reproduction) may not translate into changes in population growth rate^{21,23} if demographic compensation is occurring^{26,27}. Demographic compensation is a common phenomenon²⁸ whereby population-level declines in a given demographic rate are offset by increases in another demographic rate^{23,29,30}.” (LL 160-165)

Comment 11. Line 165: I don't agree that it has "not been investigated" and would consider even some of your own coauthor's work (e.g. Jenouvrier et al. 2018. Journal of Animal Ecology) to have done so. I would temper sentences like this and elaborate on them to more accurately reflect the state of the field or provide more nuance to your position.

Our answer: To tone down we have changed to "not been investigated much" (LL 179). We have revised the manuscript text carefully to tone down the sentences that were over-stating the results (also in line with suggestions from another reviewer). With regard to the study by Jenouvrier et al (2018): even though we admire this work, we do not think it does what we refer to in this sentence. This sentence states "Species characteristics are likely to also explain trait-mediated effects of climate on population growth, but this has not been investigated yet." (former LL 165) and we mean here that such species characteristics as generation time and diet can explain the trait-mediated effect of climate on population dynamics (the CZG path on our conceptual Fig. 1a). Jenouvrier et al. (2018) performs a detailed investigation of how several different traits mediate effects of climate on population dynamics, i.e. estimates several CZG paths. However, because Jenouvrier et al (2018) focuses on a single species it cannot explain the variation in CZG paths by species characteristics, and that is what we meant in this sentence. To improve the clarity of the text we have now revised it, as follows: "Such species characteristics are likely to also explain trait-mediated effects of climate on population growth, but this has not been investigated much" (LL 178-179).

Comment 12. Line 178: Can you elaborate on why?

Our answer: We have now expanded this sentence to "We expect the overall phenological responses to temperature to be stronger at higher latitudes¹, because of both the faster warming there³⁶⁻³⁸ and higher sensitivity of species to warming at higher latitudes³⁸." (LL 197-199).

Results:

Comment 13. Line 196: The use of the term "study" is misleading, would it not be more appropriate to refer to 116 time-series for example?

Our answer: We find this is rather a matter of taste and as long as the term is defined clearly (which we do now in a more extensive manner on LL 204-206: "Each study consists of time series of annual mean population phenotypic trait values and population sizes recorded for a unique combination of the species, location and trait."), it should not affect the clarity of the manuscript. Personally, we prefer to use "study" because in some instances it is easier to talk about studies than time series. For example, when describing the data, we find "Most of the studies on phenology recorded onset breeding..." reads better than "Most of the time series on phenology recorded onset breeding...". Similarly, when describing the random structure of the mixed-effects meta-analytical models, we find it is easier to talk about "using the study ID as the random effect" compared to "using the time series ID as random effect".

Comment 14. Line 204-236: This text would be better integrated into the introduction. The text also does not reference any of the other panels in Fig 1 and readers would benefit from more detail. Or for the results to be discussed in the context of this figure.

Our answer: Thank you. This part focusing on the description of our conceptual framework is rather technical and it would be usually found under “Methods” in the papers with classical structure (where Methods precede the Results). The Intro usually ends with the research questions/hypotheses but should not include such more technical and detailed explanations of the performed work. As the format of the Nature Communications does not allow Methods to come before Results and this rather technical description of our framework is essential to follow the Results, we prefer to place it just before the Results.

As suggested by the reviewer, we now incorporate the reference to other panels in Fig. 1 (LL 252, 257, 264, 265) in this section of the MS and believe it strongly improves the clarity of the text.

Comment 15. Line 244: Do you mean “found for individual studies” or interannually for a given study?

Our answer: Thank you for spotting this. We meant “found for..”, now changed accordingly.

Methods:

Comment 16. Line 394: What are the four classes? More detail, especially sample sizes, would be useful.

Our answer: Thank you, we have now extended: “Our final dataset consisted of 213 studies extracted from 73 papers. Our studies cover four vertebrate classes: birds (54 species), mammals (10 species), reptiles (7species) and fish (3 species; see Supplementary Data and Fig. 2 for study sample sizes per trait category).” (LL 517-520)

Comment 17. Line 399-402: Could you add the percentage of each taxonomic group and types of phenological events?

Our answer: We have done so now. It is on LL 226-235, as we shifted this information to earlier in the text.

Discussion:

Comment 18. Line 290-291: This is a key point that is missing from the introduction.

Our answer: We have now added the following sentence in the introduction, on LL 146-148: “Yet, such knowledge is crucial for the field of population ecology, as well as that of functional ecology, which relies heavily on the assumption that traits have direct population consequences and thus can be used as proxies to reflect community composition¹⁸.”

Comment 19. Line 303: What are the units?

Our answer: We have now added the units (standard deviations in trait per °C per degree latitude) in the text.

Comment 20. Line 520: How was phylogenetic relatedness estimated? Was a specific package used? What was the estimated value of lambda?

Our answer: See our response to your comment 1.

Comment 21. Line 564-575: This provides important context and justification that might be better suited in the introduction.

Our answer: Thank you. We have now integrated most of this text in the Introduction, LL 187-202 (and removed it here).

Comment 22. Line 256, line 440: Minor typos

Our answer: Thank you, we have now changed "nett" to "net" on former L256.

Figures & Tables:

Comment 23. Fig 1: Does the slope and the direction of the trends in b-d reflect your predictions?

Our answer: yes, for phenological responses. We have now stated that in the legend.

Comment 24. Does the order of the subplots, which I initially read as c, a, d, b, have any significance. I think figure a would be clearer if it was separated from b-d into different panels.

Our answer: The fact the other panels are located around the panel a) is because the panel a) represents the main conceptual framework and the insets detail the relations reflected by each arrow in a). We preferred to keep the original order of the panel labels. However, to separate the main conceptual framework in a) better from the insets we have now enclosed insets in boxes. Hope this improves the readability.

Figure 2:

Comment 25. The inset should be wider to better separate the two categories. As is, the three words read more like the x-axis label.

Our answer: Thank you for pointing this out, we have now replotted this figure to make the inset wider.

Comment 26. This figure is very visually appealing, but the font color makes some of the diet letters hard to read. I also did not feel the black bars were necessary as the numbers themselves are present and easily interpretable.

Our answer: Thank you. We have now chosen different colours for the diet and migratory mode. We have decided to retain the bars next to the numbers.

Comment 27. What does the diet "LC" stand for?

Our answer: Thank you for spotting, this was a typo, it should have been "C". Now corrected.

Figure 3:

Comment 28. To better illustrate the relationship between a and c, I would put panel a under panel c and have a upward pointing arrow.

Our answer: Since other reviewers were also not entirely happy with this figure and highlighted the concerns about non-conventional order of the panels (not starting with the a) as is usually done), we have now modified the figure to remove the arrows between the panels and rearranged them such that they follow the conventional order (i.e. a-c).

Figure 4:

Comment 29. Line 828-831: As written the caption makes it sound like there is a trend in panel a, but it is also not significant. This could be more clearly written to clearly reflect the statistics.

Our answer: Thank you for spotting this! We have mistakenly plotted the line as dashed and not a solid one, in fact this relation is significant (see SI Table S2 and main text LL 347-348). We have now corrected the figure.

Comment 30. Having the full terms on the axes, not the letter acronym, would make these plots easier to understand with a quick glance.

Our answer: Agreed, we have replotted the figure now with the full names of the effects. Hopefully this improved the readability.

Reviewer #4 (Remarks on code availability):

Comment 31. I looked through the GitHub repository and several of the R code files. The repo structure is clear and well organized. It also includes a README that identifies all the key files.

Looking through the code, I thought it was well formatted and would be easy to run if I had the climate data.

Our answer: Thank you for your positive feedback.

Response letter

We are glad that R1, R3, and R4 were happy with how we addressed their previous comments in our previous revised MS. In our new revision, we have addressed the remaining comments of R2, as well as accommodated the two remaining minor points of R4. All line numbers are for the manuscript version (and SI) with the "Track changes" mode activated.

REVIEWER COMMENTS

Reviewer #2 (Remarks to the Author):

Response to response

All line numbers are for track changes doc

I would like to start by thanking the authors for their efforts in addressing my comments. There has clearly been a lot of work put in particularly to improve the diversity of literature cited, to reduce the over-generalisation, and to have a better acknowledgement of the study limitations. However, there are a few conceptual and interpretation concerns that I do not feel were sufficiently addressed. I'm afraid I also have an additional request/suggestion for the supporting information that has arisen with the addition of new components. Overall, I still think this will be an important paper, but this means it is even more crucial to make sure it is as robust as possible, and no conclusions are misleading.

The key areas that I feel were not yet addressed sufficiently were (split into changes I suggest and then rationale for changes):

Justification of morphological traits considered here (Comment 8).

Proposed changes:

Add more contextual discussion on the morphological traits considered to the introduction (like L179-201 for phenology). Should include some expectations and discussion of the different ways morphological traits could show interannual variation at the population level – as in response to comment 8.

Rationale: In the response to reviewer comments the justification for including non-labile traits put forward was population level mean changing between cohorts possibly due to labile or developmental plasticity. However, these suggested mechanisms are quite different processes to the other traits considered (phenology and labile morphology e.g. body mass). Therefore, greater discussion of the morphological traits is required in the manuscript itself. At present the introduction reads like a phenology introduction – a good one for phenology – but missing

morphology. Morphology is only mentioned 3 times in the whole introduction, once to say morphology responds to climate change then to introduce the data and approach – I think `this can leave a reader questioning the inclusion of morphology, particularly the variety of traits. As an example, the authors could expand on the call for more studies in the Teplitsky and Millien (2014) paper they cite to help support the inclusion of morphology.

Our answer: Done. We have now included a paragraph in the Introduction that motivates further research on morphological responses to climate (inspired by Teplitsky & Millien 2014) and on explaining heterogeneity in such responses by species traits. This paragraph also highlights different ways morphological traits could show interannual variation at the population level (LL 196-210).

Interpretation and conclusions drawn from meta-analysis average effect (Comment 9, 10, 11).

Proposed changes 1: remove or tone-down/carefully caveat the conclusion statements e.g. ‘demonstrate that phenology mediates effects of temperature on population growth in vertebrates found across most of the globe’, L367-369 (e.g. result of meta-analysis showed non-negative population growth response on average, but highly variable across studies with many showing negative response, partly driven by type of phenological event), L378-382 (I really don’t think this part is needed – it just detracts from the other results – suggest remove), L472 (add in ‘many’ vertebrates).

Rationale: Thank you for your detailed responses here and clarifications on the meta-analysis. I think I can understand better your motivation and expectations for the meta-analysis. Here is my interpretation: you expected that you would be able to find a global effect size for trait-mediated responses to climate for phenology and morphology and that therefore the traits included and species etc were assumed to all be telling you about the same process – therefore a global effect made sense. I understand this motivation; but then we get to the results. At this point it is shown that there is actually a lot of variability in the process (probably even different processes going on) with some studies showing opposite patterns to others (for phenology and morphology). This is all very well assessed in the newly moved text in ‘Explaining among-study heterogeneity in climate effects’. My issue is then that the first paragraph of the discussion where you conclude that you ‘demonstrate that phenology mediates effects of temperature on population growth in vertebrates found across most of the globe’. I do not feel this a valid conclusion to draw. Yes, it is the result of the meta-analysis, but I think one has to consider whether it is sensible to draw such a global or average result when there are influential structuring elements missing from the model e.g. phenological trait type, which you know are

important. The conclusion could even be considered misleading making it sound like there is universal mediation, when in fact several studies show quite strong negative impacts on population size and misses all the nice nuance from the results section. I would recommend analysing the heterogeneity not reducing a global effect on something so variable, given the results found. If you wish to retain a global effect summary, I recommend it be clearly caveated as a statistical result rather than a biological conclusion. This is done well in the results already. I would like to add here that all the rest of the results from the path analysis and analysis of heterogeneity are already interesting and important enough, I do not feel collapsing to a global effect adds anything scientifically.

Our answer: Done. We have now toned down the statement at the beginning of the discussion, it reads as follows: “Here, by using a large set of species and studies, we demonstrate that the phenology-mediated effect of temperature on population growth rate was on average non-negative across studies. Thus, spring phenology facilitates adaptive population growth responses to temperature in many vertebrate species found across much of the globe. However, we found high heterogeneity across studies, with a substantial number showing a negative maladaptive response, partly driven by type of phenological event.” We also deleted the former sentence on LL378-382 and added “many” in front of “vertebrates”, as suggested by the reviewer.

This way we highlight heterogeneity across the studies. But we still report our main finding that phenology-mediated effect of temperature on population growth rate was non-negative across the studies. We disagree with the reviewer that if one finds variability in a process it is no longer possible to summarize the overall effect anymore. That is how scientific process works more generally, and meta-analysis in particular: we describe summary statistics and perform hypothesis testing on means. Besides, we can never expect all species and all studies to show exactly the same response. It is not the aim of the meta-analysis to reveal that all species respond exactly in the same way. Instead, the aim and the advantage of the meta-analysis is to borrow strength from studies that may have low power (and non-significant effects) if considered alone to infer about a general, across-study effect.

Proposed changes 2: Elevate the prominence of the meta-analysis as a key aim of the study (at end of intro), including discussion of how you think the ‘global effect’ can actually be interpreted biologically. Particularly in the case when effects of different sign are found between studies. But avoiding overgeneralised or misleading conclusions (where important and explainable variability is missed). Change terminology around ‘average or on average’ to better reflect the meta-analytical origin of the number e.g. global effect or meta-analysis effect or meta-analysis

average.

Rationale: In the response the authors mention the importance of the average being a meta-analysis average. However, in the text, it is referred to it as 'on average' or just 'average' several times (e.g. L279, 314, 379, 1030). This explains my previous use of such language too. If the meta-analysis part is important, this should be elevated in the text so much misreadings are less likely. It should also be clearer in the text that a meta-analysis is a main aim of the paper. Currently that can be somewhat missed as the introduction positions the paper as if the main aim is a path analysis. Meta-analysis is not even mentioned there. This also happens in the results where the cross-study effect ZG is only mentioned in one sentence (and not made clear this is from the meta-analysis), but there is a large discussion of the binomial test results (paragraph at L307). This binomial test was what I referred to previously as the analysis of majority trends in comments 9 and 11, I was not suggesting vote counting. I had meant the results of the binomial test felt more informative than a global effect, but I now see the value of the heterogeneity results.

Our answer: Done. As suggested, we now explicitly state in the Introduction that “To assess how general such effects are across species, comparative analyses such as a meta-analysis are especially valuable.” (LL 162-163). We also mention that meta-analysis is one of the two methods used in this study: “We used path analysis to (i) quantify the trait-mediated effects of climate on population growth rate (Fig. 1a) and meta-analysis to (ii) assess how general those trait-mediated effects are across the studies and to (iii) identify which type of species and regions exhibit the strongest trait-mediated effects of climate on population growth rate (using migratory mode, diet and generation time as explanatory species characteristics, and latitude to explain geographic variation among locations).” (LL 220-226)

Regarding the terminology, we have used “across-study effect”, which is an established term in meta-analysis to refer to what the reviewer calls “global effect size” (also known as “mean effect size” or “combined effect size”). When referring to the results of meta-analysis we also often have used “across studies”, which we think best (and in a most accessible and non-technical way) captures the result of meta-analysis. We have carefully read over the manuscript and in several places where we used “on average” changed it to “across studies”, to avoid a potential misunderstanding that the reviewer mentions.

We also introduce the possibility when effects of different sign are found across the studies (LL 291-295), see also a response to your next comment.

Proposed changes 3: following from Comment 10, when introducing the expectation, it would also be good to acknowledge that there are several ways to end up with a

non-negative meta-analysis average, not just from all studies being adaptive to different degrees. It could arise from some highly adaptive results and others showing mild maladaptation, but still with a global effect that is non-negative (this is an extreme example to try and illustrate the point).

Rationale: Included in the proposed change.

Our answer: Done. We addressed this by including the text “Note that non-negative CZG on average across studies can be obtained not only when majority of the studies show positive or close-to-zero CZG but also when effects of different sign are found across studies, e.g. some studies showing positive CZG (adaptive responses) and others showing slightly negative CZG (maladaptive responses).” on LL 291-295.

Accidental under-generalisation that has arisen from my suggestion L367

(Comment 14) L367 – my previous suggestion to limit this to spring phenology was a mistake given the new information on the type of phenological events included being only 75% spring. But overall would consider removing this whole sentence for reasons given above.

Our answer: This sentence is revised in response to the respective comment of the reviewer above (“proposed changes 1” above).

Comment 17. Thank you for this change but the reference to the equator is still in the abstract, could that be changed as well?

Our answer: Done. We have revised that sentence in the abstract as follows “At lower latitudes, temperature had weaker effects on phenology but stronger direct negative effects on population growth, likely because these populations are less capable of tracking climate via plasticity.” (LL115-118)

Comment 27, not sure I fully understand the response. I’ve tried here to explain my point and interpretation of the response a bit more with a specific improvement suggestion.

Proposed changes: to acknowledge that any identified drivers will be proxies so do not need full biological realism (impossible really with an absolute window).

Rationale: To my knowledge of how sliding windows work, they start at a reference day and search back in time i.e. earlier than the reference. An example would be a reference day of 20th May then a 1 year search would be from 20th May Year1 – 20th May Year0. If the event was for example, breeding timing and recorded breeding was 24th May to 5th June across years, this would all be ok as the

identified window was occurring before the event in all years. I find a bit of a problem with a temperature window for phenology occurring after the event has happened, which could occur if you set the reference day to the latest observation e.g. 5th June in the previous example. It could select a window of first week of June when most events have happened in May and therefore biologically could not have had any influence on those events. A causal relationship is impossible. However, this debate does not really matter that much, as all absolute windows will be proxies anyway and unlikely to be causal. However it is worth clarifying as sliding windows are often assumed as representing causal cues in phenology and could be misinterpreted in that field. For morphology, taking the latest date could make more sense, though, having the driver occurring before morphology is measured does still make sense. My key point is a driver window later than an event or measurement cannot influence it, one earlier can. But do let me know if I have misunderstood this.

Our answer: We try to explain here better why we have chosen to use as a reference day the latest day in the year when the phenological event was observed over the study period. For this example, let us use as a phenological trait of interest a median (across sampled nests, within a year) egg laying date. Let us consider, for simplicity, a time series of 10 years. Let us say that across these 10 years the latest the median date was observed was 5th of June. And the earliest recorded median egg laying date (across 10 years) was 10th of May. In our analyses we use the 5th of June as a reference day and look backwards (up to 2 years) to identify a climatic window that best predicts the egg laying date across all years. So, the climatic window always precedes the latest day (across the time period) when phenotypic trait was measured because this climatic window is assumed to trigger the trait of interest. We represent this schematically in the figure 1 below (by using somewhat different egg laying dates, typical of passerines in Europe). By using the latest date over the study period when phenological trait was recorded we avoid missing or misestimating the relevant climatic window (see Fig. 1). We hope this clarifies the confusion around the reference day. To clarify this in the text, we state that “By using the latest day in the year when the phenological trait was observed over the study period we assure that no relevant windows will be missed (Supplementary Fig. S15)” (LL 588-590). We also say that the identified windows may reflect proxies of the true underlying biological mechanisms (LL 598-599). We also include a (somewhat modified) figure presented here as Supplementary Fig. S15 in the Supplementary Information.

Figure 1. Schematic representation of the sliding window analysis and the rationale for using as a reference day the latest date when the phenological event was observed or morphological trait was measured in the year over the study period. In this hypothetical study phenological trait (median egg laying date) of passerine birds was recorded over 10 years, from 2000 to 2009. The example focuses on the year 2001 and the orange and the green vertical lines depict, respectively, the latest and the earliest recorded median egg laying date over the 10-year study period. The grey double-headed arrows show (a selection) of tested windows. The blue double-headed horizontal arrow shows the best selected climatic window. Had we used as the reference day the earliest median egg laying date recorded over the 10 years, we would have failed to identify this climatic window (as any climatic window when using this reference day must stop at the latest by the time point depicted with green vertical line). By using the latest day when the even was observed as our reference day we thus try to strike a balance between false negatives and false positives.

New comment: Structure of Methods and Supplementary Material. Having re-read the Methods and Supporting information and possibly because of all the additions in the revision, I found some of the styles and structure hard to follow. I will try and be as specific as I can.

Proposed changes: more thought about the structure of the supporting information would be really helpful, I would recommend structuring to mirror the mentions in the main text or a standard paper i.e. intro related, methods related, results related, etc. I found the current structure with so much material and so many different models to be quite confusing to follow.

Our answer: In our supporting information the numbering of the figures and tables follows the order in which they are mentioned in the main text. To our knowledge that is a common practice and the figures and tables in the main

text actually must be cited in that consequent order. To help the reader navigate the supplementary information, we split the materials into Supplementary Figures, Supplementary Tables, Supplementary Methods, Supplementary Results and provide table of contents at the beginning of the SI with respective page numbers.

Supplementary Information is not meant to be read as a stand-alone document and therefore we do not think that re-structuring it to “mirror a standard paper into intro-related, methods-related...” sections is useful. Restructuring in this way will also result in the numbering of the Tables and Figures that will not be sequential when mentioned in the main text.

Therefore, we have kept the original structure of SI but are willing to restructure it differently if such a change is deemed useful by the editor.

Explain more clearly how many meta-analytical models are fitted (maybe a table of them all) and justify the model structures e.g. why not include all important drivers throughout? Also, explain which model has led to the ‘global effect’ quoted and how any other model structures impact this quantity (global effect).

Our answer: Answered in detail below, where reviewer 2 goes into this point in more detail. Briefly: only two main model types were fitted, addressing two main research questions of this study. As suggested by the reviewer, we include a table of all models as Supplementary Table S8.

Re-name the sensitivity section, include how these models were run (as this is the methods not results) and move results to supporting information.

Our answer: See the detailed response below. We moved this section to the Results section, as its contents indeed better fit there.

Be clearer about the population size impact, or better yet, just include this variable in all models. If not the latter, justify why it is not included. Also correct Figure S18 as figure does not match interpretation text below i.e. graph says adding population size makes the effects significant, text says opposite.

Our answer: Population size was included as an explanatory variable of population growth rate in the structural equation models fitted to each study (see conceptual Fig. 1, LL 632-634 and LL 638-641, and Fig. S18 that shows the methodological workflow of this study). Thus, it is included already in all models and this was explained already in the previous version of the manuscript.

See below for a more detailed answer to the comment on Figure S18.

Main areas of confusion:

1. How many models? It is not clear to me how many meta-analytical models were actually fitted and which are interpreted in the main text. In the methods in main text, it seems like an initial one with effects of study, location, and species as well as phylogenetic relatedness and also accounting for probability of being spurious and climate data quality, then one without phylogeny, then a third with the variables hypothesised to impact phenological responses (but not done for morphology?), then a fourth to test the phenological trait variable. I have a question as to why there are so many different models. Could the full model with variables hypothesized to be important not just have been used from the start and phylogeny influence checked then? It is not clear currently why so many model variants are being used in different parts and what impact this has on the results. Apologies if I have missed this, if I have, please just elevate mentions. Currently a reader can be left questioning reasons for choices.

Our answer: To clarify this point we have now included the Supplementary Table S8 in SI that provides a detailed overview of the fitted meta-analytical models and refer to this table on LL 773-774.

In more detail, all fitted models are described in the Methods section of the manuscript. Also, our former Supplementary Figure S10 (now Supplementary Fig. S18) schematically represents the methodological workflow and shows separately the analyses that were run for each single study and those that were run across studies. For each single study we fitted structural equation model, to estimate for each study the trait-mediated effect of climate on population growth rates (and other paths visualised in our conceptual Fig. 1). These estimates were then used as response variables in a meta-analysis (while propagating the uncertainty). We fitted two types of meta-analytical models, to address two main questions of our study (as highlighted on LL 222-226): “We used ... meta-analysis to (ii) assess how general those trait-mediated effects are across the studies and to (iii) identify which type of species and regions exhibit the strongest trait-mediated effects of climate on population growth rate (using migratory mode, diet and generation time as explanatory species characteristics, and latitude to explain geographic variation among locations).”

First, to assess how general the responses are across the studies (aim “ii” above), we fitted a meta-analytical model to assess the across-study effect size. This model controlled for the effects of study, location, and species (included as random effects), and for phylogenetic relatedness. We fitted these models separately for each combination of the phenotypic trait type (phenology, morphology) and climate variable (temperature and precipitation).

One model was fitted per path shown in our conceptual diagram in Fig. 1 (CZ, ZG, PG, CG, CZG, total CG), using the path as a response variable. For the models that had as response variables the paths that included climate (e.g. CZ, CZG), we also included probability of the climate effect being spurious and climate data quality as covariates. We then fitted exactly the same model structure but without phylogenetic relatedness and compared both models using AIC, to avoid overfitting. In the main text we only report the results of the models for temperature-phenology combinations, while the results for three other combinations of traits and climate variable (precipitation-phenology, temperature-morphology and precipitation-morphology) are only reported in SI because traits were insensitive to climate in these combinations, as reported in the main text (LL 298-301 and LL 333-341).

Second, to assess how species characteristics and latitude affect trait-mediated effects of climate on population growth rate (aim “iii” above), we have fitted a meta-analysis with the hypothesised species characteristics and the latitude as predictors. These models included probability of the climate effect being spurious as a covariate and were fitted only for studies focusing on phenological traits and effects of temperature, because only for this trait-climate combination the likelihood that the identified climate windows were spurious was low across the studies. And, these models were fitted for the key paths of interest from our conceptual diagram in Fig. 1: CZ, ZG, CZG, and CG. Finally, to assess how well heterogeneity in phenological responses to temperature can be explained by a specific phenological trait measured, we run a meta-analysis with the specific phenological trait type as a predictor.

So, there are no different model variants used, as the reviewer seems to suggest. There are really two types of models that are needed to address our two research questions.

While constructing the table we have realised that we have not always systematically reported the effects for one of the considered paths, CG. We therefore updated respective figures to visualise CG along with other paths (this concerns several figures in SI).

2. Sensitivity analysis/Missing variables. This doesn't feel like the correct title for this section. It seems more like it is justifying not including certain variables in all models rather. I also typically consider sensitivity to assess impact of elements of the final model, not additional things that were not included. It was also a bit confused including both a missed variable and justification of not removing possibly spurious relationships. Furthermore, it reads as results not methods. It does not actually say how these models were run, just saying what their results were. It seemed a bit confusing here and should be in results? Or referenced in results but in supporting

information and just referenced at relevant places such as L676-677 when the variable accounting for the probability of being spurious is mentioned.

Our answer: Exactly as the reviewer describes, we here assessed sensitivity of the meta-analyses results to the predictors that were included in the model. See below for the reviewer's comment on population size – it was included in all structural equation models (SEMs from now on) by default and we conducted the sensitivity analysis to test how excluding this variable from our SEMs affects the findings. In a similar way, we here test what is the effect of including potentially spurious relationships (by using $P_{\Delta AICc}$ as a covariate) on our findings. We did not exclude this variable from the analyses, but rather tested how it affected the estimated across-study effect sizes. Perhaps the confusion stems from the original wording of this section, so we have revised it now to read as follows: “However, the results of the meta-analyses were not qualitatively affected by inclusion/exclusion of (i) the less supported climate signals in the models and (ii) population size as a covariate (Supplementary Figs S8-S11)” (LL 390-392).

We agree with the reviewer that this information belongs rather to the Results than Methods section and have now shifted it to the Results part. Shifting it to the Results led to the change in the numbering of some of the figures in SI.

3. Population size. It is clear that population size is highly important for G, so why was this variable not just included in the models interpreted in the text rather than being pushed to the supplementary? It seems that inclusion or not impacted significance of some effects, which is important but shrugged off here. I do not think it is currently helped that wording in the Supporting Information does not match the graph i.e. graph says adding population size makes the effects significant, text says opposite (Figure S18). But I wonder if there is an effect you know even a priori will be important, why not include it throughout? If there is a reason, it should be included. So, either switch the model or add a concrete reason why population size is omitted.

Our answer: Population size was already included in all SEMs described in the main text, as the Reviewer asks for. This was already explained in the conceptual Fig. 1, stated on LL 632-634 and LL 638-641 (in the section “Trait-mediated effects of climate on G”) and shown in the figure S18 that depicts the methodological workflow. This analysis intended to assess, exactly as the reviewer suggests, what would be the impact on the conclusions drawn from the meta-analysis had we **not** included population size in our SEMs. So indeed, we expected the effect of population size to be important and we did include it in all structural equation models fitted to each study.

We disagree with the suggestion of the reviewer that the wording in the SI does not match the figure: the figure says that if we exclude the population

size from the SEMs (depicted with circles on Fig. S11 (former Fig. S18); the crosses show results when population size is included in SEMs, our default approach reported in the main text) then the across-study effect of the meta-analysis becomes significant for the path ZG (precipitation effect).

So, the finding becomes significant if we exclude population size from the analyses. And that is exactly what the legend to this figure says: “The exclusion of population size from path analysis does not affect the meta-analysis’ results qualitatively, except that for studies on phenology (b) omitting the population size (DD on the figure legend) results in one coefficient being significant: the effect of trait on population growth rate when assessing the effects of precipitation (ZG, compare dark green circle vs light green cross).”

4. Structure of the supporting information. I would like to suggest structuring as ‘Supplementary Methods’, then Results, then Figures, or interspersing the figures where relevant. The current structure makes some things hard to understand. Some examples: Figure S7 – should this not be first in figures as it refers to an initial step in the analysis? Also, Figure S10 possibly could go earlier. Table S7 appears between several with results which feels a bit odd, could be first table as it is contextual or about data not results. Moving methods before figures and tables of results would make them much easier to understand, e.g. Table S8.

Our answer: See our previous answer above, where the reviewer made the same comment.

Small details: Figure S5 mentions different sample size as SEMs for two taxon did not converge, was this also the case for the main analyses in main text. I might have missed it but this did not seem to be mentioned in relation to those analyses but should be.

Our answer: This was shown in the Supplementary Fig. S14 (now Supplementary Fig. S7). We have now also added a sentence to the Methods section (LL 662-667) stating this: “SEMs failed to converge for several studies resulting in sample sizes for meta-analyses being somewhat lower than the number of originally retrieved studies: 93 studies for phenological responses to temperature, 95 studies for phenological responses to precipitation, 109 studies for morphological responses to temperature and 115 studies for morphological responses to precipitation (Supplementary Fig. S7).”

Figure S8 – don’t jitter if you want to show comparability between two metrics. Also, why only for 35 studies? And how were they chosen?

Our answer: Thank you! Done. We do not jitter in the vertical direction anymore and use a different symbol (an empty one), that enables visualising

the symbols even if they overlap. We have added the following rationale of our choice of 35 studies to the legend of the Figure S16 (former Fig. S8): “We chose 35 studies as a compromise between running time and having a broad enough and representative sample of all studies. Running sliding window analysis for a study with duration of 30 years using daily resolution on a computing cluster required around one day while the same analyses with weekly resolution required two-three hours. We therefore have chosen 35 studies that constitute >1/3 of phenological studies and focus on birds only (as birds represent 87% of all studies)”.

Figure S17 – what are the numbers? Are they sample size? It seems sample sizes might be more variable than has been mentioned, could this please be addressed.

Our answer: These are indeed sample sizes, we have now added a sentence to the legend of this figure (and other similar figures) clarifying this. See the response to your other comment above: we have also added a text in the Methods section clarifying that sample sizes for meta-analyses are somewhat lower (as indicated in Supplementary Fig. S7) because SEMs failed to converge for several studies: LL 662-667.

Section ‘testing for non-linearity in relations’ is very repetitive.

Our answer: We have revised this section to avoid repetitions.

Reviewer #2 (Remarks on code availability):

As before, all good.

Reviewer #3 (Remarks to the Author):

Reviewer #4 (Remarks to the Author):

Thank you very much for the extensive revision of the paper. The additional citations, elaboration in the introduction and details in the methods greatly improved the manuscript and addressed my comments.

Line 508-516: These sentences could be reorganized to improve the flow of ideas and reduce the repetitive phrasing.

Our answer: Done, we have revised this section to improve the flow.

Line 559-560: Can you be clearer as to why you think this is unfortunate? I assume you mean it is unfortunate that we are still biased in where most ecological time-series data are collected, but perhaps you mean something else about these studies specifically.

Our answer: Indeed, this is what we meant. To state it unambiguously, we have revised this sentence as follows: “The majority of data stem from the northern hemisphere, particularly Europe and North America (Fig. 2), as is the case with many recent global meta-analyses in ecology that are unfortunately biased in where time-series data are collected ^{1,19,48}.”

Response letter

REVIEWERS' COMMENTS

Reviewer #2 (Remarks to the Author):

I would like to thank the authors for their careful consideration of my comments and additions/changes to the manuscript. In particular, the wording of 'across-study effect', the tweaking of the conclusion statements, the clarification of models (Supplementary Table S8), and addition of motivation for morphology in the intro.

I also appreciate the clarification on Figure S11, I can now see that the figure and text do match. Though it seems the Figure has changed from the previous version so now only one effect is made significant by the removal of population size. I think it was the lack of circle for the PG effect that confused me before, I assume this is because it cannot be estimated without population size (which makes sense), but perhaps this could be stated in the legend just to make sure it cannot be missed by readers.

Our answer: We have now revised the legend to this figure to be as explicit as possible about the fact that analyses were run either with or without the population size to test how sensitive our conclusions are: "The across-study effect sizes for each relation in the path diagram used in the path analysis (Fig. 1, main text) are shown for analyses with (a cross symbol for 'Yes') and without (an empty circle symbol for 'No') the effect of population size on population growth rate (signified as DD on the figure legend)".

I also appreciate the time spent to explain the choice of reference day for the sliding window and the new figure.

I do think that the figure the authors presented in their response should be the one in the text. Showing the earliest date as well as the latest is very important, which I explain below. While I would like to emphasize that I think that the author's response is sufficient now for publication (thank you for this), I would still like to justify the use of the figure in the response. To do this, I have edited the authors' nice figure (attached as I cannot upload into this text box). What I have done is cut out the window around Oct (the shortest window in the figure) and move it to closer to the reference day. So, this should be a possible window that could be selected as the 'best' window. I hope this now shows the biological issue, which is that this window cannot have influenced egg laying for the earliest recorded events, as the window occurs after these events. It can correlate with them, but it cannot be causal for all years. Such an outcome may not have happened in any cases in this study (I don't know) but it could by using the latest event as the reference day. The response figure

already starts to illustrate this as the 'best' window continues beyond the earliest event, but the Supplementary Figure does not. Therefore, including the schematic with the earliest recorded line will better illustrate this possibility to readers and will help correct interpretation of results, adding a possible window fully after the first date would be even better and more transparent. I would suggest a small wording change here, as a window that occurs after earlier events cannot be biologically 'relevant' in those years — though it may still correlate statistically. Thank you for the involved discussion on this, I do hope you will consider this point in any final adjustments.

Our response: We thank the reviewer for the detailed explanation. We have now included the exact figure that was previously included in the response letter as the Supplementary Figure S15. We also revised the figure legend to highlight the possibility of the phenomenon that is described by the reviewer, in some years (when egg laying could start earlier than the selected climatic window). The legend now reads: "Schematic representation of the sliding window analysis and the rationale for using the latest date when the phenological event was observed or morphological trait was measured as a reference day. In this hypothetical study the phenological trait (median egg laying date) of passerine birds was recorded over 10 years, from 2000 to 2009. The example focuses on the year 2001 and the orange and the green vertical lines depict, respectively, the latest and the earliest recorded median egg laying date over the 10-year study period. The grey double-headed arrows show (a selection) of tested windows. The blue double-headed horizontal arrow shows the best selected climatic window. Had we used the earliest median egg laying date recorded over the 10 years as the reference day, we would have failed to identify this climatic window (as any climatic window when using this reference day must stop at the latest time point depicted with green vertical line). Note that if the best selected window occurs after the earliest recorded egg laying date, it may not be relevant for egg laying dates observed in each year during the study (as in some years it may potentially occur after the egg laying started). By using the latest day when the event was observed as our reference day, we thus try to strike a balance between false negatives and false positives."

Reviewer #2 (Remarks on code availability):

Didn't re-check this time but assume still good as before.

Reviewer #3 (Remarks to the Author):
